# Opn5L1 is a retinal receptor that behaves as a reverse and self-regenerating photoreceptor

Keita Sato [1], Takahiro Yamashita[2], Hideyo Ohuchi[1], Atsuko Takeuchi[3], Hitoshi Gotoh[4], Katsuhiko Ono[4], Misao Mizuno[5], Yasuhisa Mizutani[5], Sayuri Tomonari [6], Kazumi Sakai[2], Yasushi Imamoto[2], Akimori Wada[7] & Yoshinori Shichida[2,8]

Most opsins are G protein-coupled receptors that utilize retinal both as a ligand and as a chromophore. Opsins' main established mechanism is light-triggered activation through retinal 11-*cis*-to-all-*trans* photoisomerization. Here we report a vertebrate non-visual opsin that functions as a Gi-coupled retinal receptor that is deactivated by light and can thermally self-regenerate. This opsin, Opn5L1, binds exclusively to all-*trans*-retinal. More interestingly, the light-induced deactivation through retinal *trans*-to-*cis* isomerization is followed by formation of a covalent adduct between retinal and a nearby cysteine, which breaks the retinal-conjugated double bond system, probably at the $C_{11}$ position, resulting in thermal re-isomerization to all-*trans*-retinal. Thus, Opn5L1 acts as a reverse photoreceptor. We conclude that, like vertebrate rhodopsin, Opn5L1 is a unidirectional optical switch optimized from an ancestral bidirectional optical switch, such as invertebrate rhodopsin, to increase the S/N ratio of the signal transduction, although the direction of optimization is opposite to that of vertebrate rhodopsin.

[1] Department of Cytology and Histology, Okayama University Graduate School of Medicine, Dentistry and Pharmaceutical Sciences, Okayama 700-8558, Japan. [2] Department of Biophysics, Graduate School of Science, Kyoto University, Kyoto 606-8502, Japan. [3] Division of Analytical Laboratory, Kobe Pharmaceutical University, Kobe 658-8558, Japan. [4] Department of Biology, Kyoto Prefectural University of Medicine, Kyoto 603-8334, Japan. [5] Department of Chemistry, Graduate School of Science, Osaka University, Osaka 560-0043, Japan. [6] Division of Chemical and Physical Analyses, Center for Technical Support, Institute of Technology and Science, Tokushima University, Tokushima 770-8506, Japan. [7] Department of Organic Chemistry for Life Science, Kobe Pharmaceutical University, Kobe 658-8558, Japan. [8] Research Organization for Science and Technology, Ritsumeikan University, Kusatsu, Shiga 525-8577, Japan. Correspondence and requests for materials should be addressed to Y.S. (email: shichida@rh.biophys.kyoto-u.ac.jp)

Most animals express light-sensing receptor proteins called opsins that form the molecular basis of visual and non-visual photoreception[1–3]. Most opsins are G protein-coupled receptors (GPCRs) that have a seven-transmembrane α-helical structure and contain the vitamin A-derivative retinal as an intrinsic ligand and light-absorbing chromophore[4]. Recent progress in cloning and sequencing technologies have helped identify more than 20,000 opsins from various animals, and to classify them into at least seven phylogenetic groups[5–7]. In addition, based on their molecular functions, these opsin groups can be divided into two categories that function either as GPCRs by light-dependent G protein activation or as retinal photoisomerases by producing 11-*cis*-retinal from the all-*trans* form[2].

All the GPCR-type opsins characterized so far bind to 11-*cis*-retinal in their resting state. From a pharmacological perspective, 11-*cis*-retinal can be considered an inverse agonist. Such opsins are converted to the active state by the isomerization of 11-*cis*-retinal to the agonist all-*trans*-retinal following the absorption of light[8]. Most GPCR-type opsins function as bistable (bidirectional) photoreceptors whose resting and G protein-activating states are interconvertible in a light-dependent manner[9]. These opsins can be activated or inactivated under appropriate light conditions.

In contrast, vertebrate rhodopsin evolved as a monostable (unidirectional) photoreceptor that exclusively exhibits a photoreaction that converts it to the G protein-activating state, and lacks a photoreaction to induce its reversion to the resting state. An important consequence of this unidirectional photoreaction is improved G protein activation[10].

Opn5 was first identified in the human and mouse genomes, and is expressed in the eye, brain, spinal cord, and testis of both species[11]. Subsequent investigations have revealed at least three phylogenetic subgroups[12] (Supplementary Fig. 1). The best studied is Opn5, or mammalian-type Opn5 (Opn5m), a Gi-coupled UV light-sensitive bistable opsin[13] responsible for photoperiodic induction of testicular growth in birds[14] and for photoentrainment of local circadian oscillators in the mammalian retina and cornea[15]. In addition, Opn5L2, which was first identified in the chicken genome as Opn5-like opsin 2, also functions as a Gi-coupled UV light-sensitive bistable opsin[12,16]. Here, we show that Opn5L1, or Opn5-like opsin 1[12], has considerably different molecular properties from those of known opsins, including Opn5m and Opn5L2. That is, Opn5L1 functions as a Gi-coupled retinal receptor, since it binds exclusively to all-*trans*-retinal to form an active state. However, when it absorbs a photon, it is inactivated by *trans*-to-*cis* isomerization of the retinal followed by formation of a covalent adduct of the retinal with a nearby cysteine residue. The adduct formation breaks the retinal-conjugated double bond system, probably at position $C_{11}$, which enables the $C_{11}-C_{12}$ single bond to thermally re-isomerize to the trans-conformation. Thus, Opn5L1 acts as a reverse photoreceptor that loses its activity upon photon absorption and is capable of slow self-regeneration from the stable inactive state. We describe in detail the properties of Opn5L1 and discuss the significance of Opn5L1 during the course of opsin molecular evolution.

## Results

**Molecular characteristics of Opn5L1.** The *Opn5L1* gene was first isolated from the chicken genome[12] and can be found in various vertebrate species ranging from cartilaginous fish to sauropsids, but not in mammals. We selected chicken *Opn5L1* as the main subject of our study due to its high yield when expressed recombinantly. To further improve the yield of recombinant protein, we replaced the N- and C-termini of chicken Opn5L1 with those of *Xenopus tropicalis* Opn5m, which belongs to another group of Opn5 proteins with higher yields (Supplementary Fig. 2). It should be noted that Opn5L1 bearing native N- and C- termini showed molecular characteristics indistinguishable from those of Opn5L1 with the modified N- and C-termini (Supplementary Fig. 3a−d). In this study, we refer to Opn5L1 having the modified N-terminus and modified N- and C-temini as Opn5L1N and Opn5L1NC, respectively (Supplementary Fig. 2).

As all of the Opn5 proteins studied thus far have been photosensitive proteins that use 11-*cis*-retinal as their chromophore[13,16–18], we first tried to regenerate recombinant Opn5L1NC with 11-*cis*-retinal. A suspension of HEK293T cells transfected with Opn5L1NC was incubated with 10 μM 11-*cis*-retinal at 4 °C for 8 h. Following detergent-solubilization and purification, the absorption maximum of Opn5L1NC was ~510 nm (Fig. 1a). However, analysis of the retinal configuration showed that the purified protein contained predominantly all-*trans*-retinal, even though it was originally incubated with 11-*cis*-retinal (Fig. 1b). This strongly suggests that Opn5L1 does not bind 11-*cis*- but instead binds all-*trans*-retinal, which could be formed through isomerization of 11-*cis*-retinal catalyzed enzymatically by intrinsic retinoid processing machinery in cultured cells[19] or non-enzymatically by lipids[20] or nucleophiles[21] present in cell suspensions during the incubation. We verified that Opn5L1NC can indeed be regenerated by all-*trans*-retinal, and the resulting absorption spectrum was identical to that obtained following incubation with 11-*cis*-retinal (Fig. 1a). The chromophore extracted from this sample was also shown to have the all-*trans* configuration (Fig. 1b). Therefore, we concluded that Opn5L1NC directly incorporates all-*trans*-retinal and not 11-*cis*-retinal. In the following experiments, all-*trans*-retinal was added directly to cultured cells used for preparation of all-*trans*-retinal-bound Opn5L1NC to increase the yield of the recombinant protein.

Since all-*trans*-retinal acts as an agonist for all known animal opsins except for photoisomerases[1,2], we examined whether Opn5L1 bound to all-*trans*-retinal can activate G proteins. The membrane fraction containing Opn5L1N was prepared without retinal, and G protein activation was measured at various concentrations of all-*trans*-retinal (Supplementary Fig. 4a). $EC_{50}$ for retinal was estimated to be $6.2 \times 10^{-7}$ M. Thus, assuming that the retinal concentration in the chicken tissues is similar to that in mouse ($10^{-6}–10^{-8}$ M)[22,23], Opn5L1 could act as a retinal sensor in the chicken tissues. Additionally, we also measured activation of G protein by purified Opn5L1N (Fig. 1c). The results showed that Opn5L1N efficiently activated Gi-type G protein in the dark, indicating that Opn5L1 is a retinal receptor with all-*trans*-retinal as an agonist (Fig. 1c and Supplementary Fig. 4). Moreover, we attempted to determine if Opn5L1 is an all-*trans*-retinoid receptor. We did not detect any G protein activation after adding all-*trans*-retinol or retinoic acid to the membrane fraction containing the Opn5L1N apoprotein, which indicates that Opn5L1 is an all-*trans*-retinal receptor (Supplementary Fig. 4b, c). It should be noted that the retinal-free Opn5L1N apoprotein also showed no detectable G protein activity, indicating that the binding of all-*trans*-retinal to apoprotein triggers the conformational changes that activate the receptor. Furthermore, we compared the G protein activation efficiency of Opn5L1 with those of chicken Opn5m and bovine rhodopsin (Supplementary Fig. 5). The Gi activation efficiency of Opn5L1 was about fivefold higher than that of chicken Opn5m and 100-fold lower than that of bovine rhodopsin. This efficiency of Opn5L1 is similar to that of bistable opsin, which has a 50-fold lower activation efficiency than bovine rhodopsin[10].

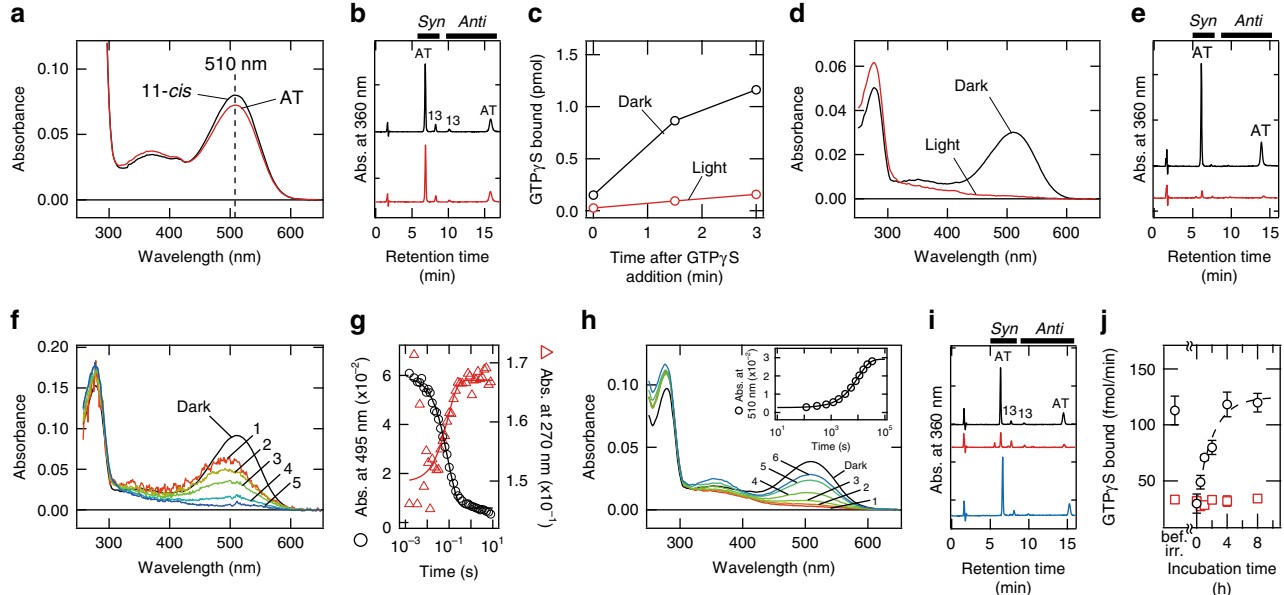

**Fig. 1** Molecular characteristics of Opn5L1. **a**, **b** Absorption spectra (**a**) and HPLC patterns of retinal chromophore (**b**) of Opn5L1NC purified after incubation with 11-*cis*- (black) or all-*trans*-retinal (red). The spectra were recorded at 10 °C. **c** G protein activation ability of Opn5L1N before (black) and after (red) irradiation with >500 nm light for 2 min. **d**, **e** Absorption spectra (**d**), and HPLC patterns of retinal chromophore (**e**) of Opn5L1NC before (black) and after (red) irradiation with >500 nm light for 2 min. The spectra were recorded at 10 °C. **f** Absorption spectra before (Dark) and 0.0015, 0.017, 0.05, 0.16, and 2.1 s (curves 1−5, respectively) after flash irradiation with >500 nm light at 37 °C. **g** The absorbance at 495 nm (black circles) and 270 nm (red triangles) plotted against the time after flash irradiation. The time profiles were fitted by single exponential functions $y = y_0 - b \times \exp(-t/\tau)$ with the same time constant (solid curves, $y = 0.0059 - 0.054 \times \exp(t/0.089)$ for 495 nm, and $y = 0.167 - 0.017 \times \exp(t/0.089)$ for 270 nm, $\tau = 0.089$ s). **h** Absorption spectra before (Dark) and 0, 11, 32, 92, 447, and 557 min (curves 1−6, respectively) after irradiation with >500 nm light for 2 min at 37 °C. (inset) The absorbance at 510 nm plotted against the time after irradiation (black circles). The data were fitted by single exponential function $y = y_0 - b \times \exp(-t/\tau)$ (solid curve, $y = 0.029 - 0.026 \times \exp(t/10500)$, $\tau = 1.1 \times 10^4$ s). **i** HPLC patterns of retinaloximes extracted before (black) and 2 min (red) or 11 h (cyan) after light irradiation. **j** Recovery of G protein activation efficiency of Opn5L1N after light irradiation. G protein activation efficiencies estimated with (red squares) and without (black circles) additional light irradiation were plotted against incubation time. Data points represent mean values ± s.d. ($n = 3$). Kinetics of the increase of G protein activation efficiency were fitted by a single exponential function (broken line) with a time constant of $7.5 \times 10^3$ s

Since Opn5L1 reconstituted with all-*trans*-retinal exhibits absorbance in the visible region, we irradiated it with visible light and found that the irradiation suppressed G protein activation (Fig. 1c). Thus, Opn5L1 can act as a reverse photoreceptor that loses its activity upon photon absorption.

Importantly, irradiation caused the complete cancellation of absorbance in the visible and near-UV regions, concomitant with a small increase in absorbance at 270 nm (Fig. 1d). Furthermore, HPLC analysis of the extract using hydroxylamine and organic solvent showed that retinal isomers could not be extracted from the irradiated sample (Fig. 1e).

To elucidate the Opn5L1 photoreaction, we performed time-resolved spectrophotometric experiments using purified Opn5L1NC. First, flash photolysis experiments indicated that a 500-nm intermediate state with a retinylidene-protonated Schiff base chromophore preceded formation of the 270-nm product (Fig. 1f, g). Additionally, we observed that the 270-nm product was gradually converted to a 510-nm state with a time constant of about 3 h at 37 °C (Fig. 1h). HPLC analysis of the retinal chromophore showed that this 510-nm state is the dark state that contains all-*trans*-retinal (Fig. 1i). Furthermore, G protein activation assays using Opn5L1N showed that the activity was also recovered by incubation at 37 °C after light irradiation (Fig. 1j). It should be noted that the addition of exogenous all-*trans*-retinal did not accelerate the recovery of the 510-nm state (Supplementary Fig. 6), which indicates that the recovery of the original dark state does not occur by the uptake of exogenous all-*trans*-retinal. In other words, the 270-nm product reverted to the original dark state.

**Reaction mechanism of Opn5L1.** Since the absorption maximum (270 nm) of the metastable product formed by the irradiation of Opn5L1 is similar to that of the protonated 11,12-dihydro-retinylidene Schiff base in methanol[24], we speculated that irradiation of all-*trans*-retinal in Opn5L1 should result in the formation of a retinal adduct with a nearby amino acid residue(s) after the *trans*-to-*cis* photoisomerization. To test this possibility, we first conducted experiments to verify that the 500-nm intermediate state has an 11-*cis*-retinal chromophore by using a previously established method[25]. To stabilize the intermediate state, we cooled the Opn5L1NC sample to −72 °C by immersing it in dry ice/ethanol prior to irradiation with >500 nm light to generate the intermediate state. This was followed by extraction of the retinal chromophore by adding methanol containing hydroxylamine to the sample. We were thereby able to extract the 11-*cis* isomer with a small amount of 9-*cis* isomer from the sample (Fig. 2a, b). In contrast, we did not extract the 11-*cis* isomer when methanol/hydroxylamine was added after warming the sample and incubating it for 30 min at 0 °C (Fig. 2a, b). These results clearly indicate that the 11-*cis* isomer was formed by the photoisomerization from the all-*trans* isomer at −72 °C and was subsequently converted to a retinal adduct to form the 270-nm product (Fig. 2c). It should be noted that the 9-*cis* isomer was still extracted from the sample warmed to 0 °C, indicating that it was not converted to a retinal adduct (Fig. 2c).

We went on to perform spectroscopic screening involving the mutation of amino acids in the proximity of retinal and identified Cys188 as the amino acid residue likely to be chiefly responsible for the formation of the 270-nm product (Supplementary Fig. 7).

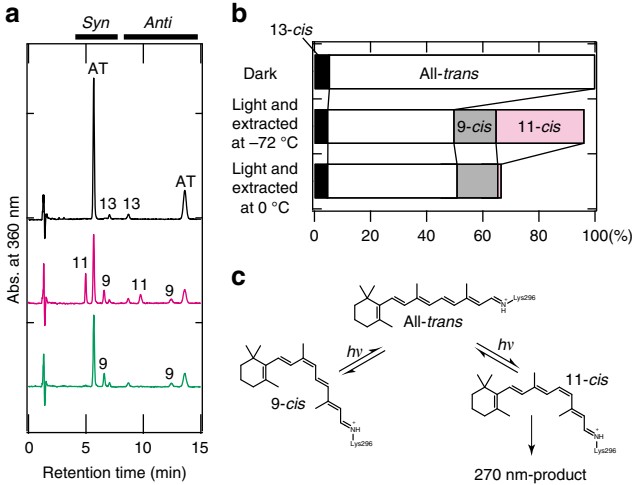

**Fig. 2** Retinal configurations in the intermediates produced by irradiation of Opn5L1 at −72 °C. **a** HPLC patterns of the retinaloximes extracted from non-irradiated Opn5L1NC sample (black), cooled at −72 °C in an ethanol/dry ice bath and irradiated with >500 nm light for 1 min (magenta), and subsequent incubation at 0 °C for 30 min (green). **b** Calculated compositions of retinal isomers in the samples based on each peak area in the chromatogram from **a** and the extinction coefficients previously reported[55]. Compositions of the retinal isomers of the dark sample (black in **a**) were 5.50 and 94.5% for the 13-cis and all-trans, respectively. Those of the sample after >500 nm light irradiation for 1 min and extraction at −72 °C (magenta in **a**) were 4.86, 45.0, 15.0, and 31.3% for the 13-cis, all-trans, 9-cis and 11-cis, respectively. Those of the sample after >500 nm light irradiation for 1 min at −72 °C, followed by extraction after incubation at 0 °C for 30 min (green in **a**) were 4.88, 46.1, 14.5, and 1.19% for the 13-cis, all-trans, 9-cis and 11-cis, respectively. **c** Reaction scheme of the chromophore of Opn5L1NC under irradiation with >500 nm light at −72 °C. It should be noted that only the 11-cis isomer was not extracted after incubation of the irradiated sample at 0 °C

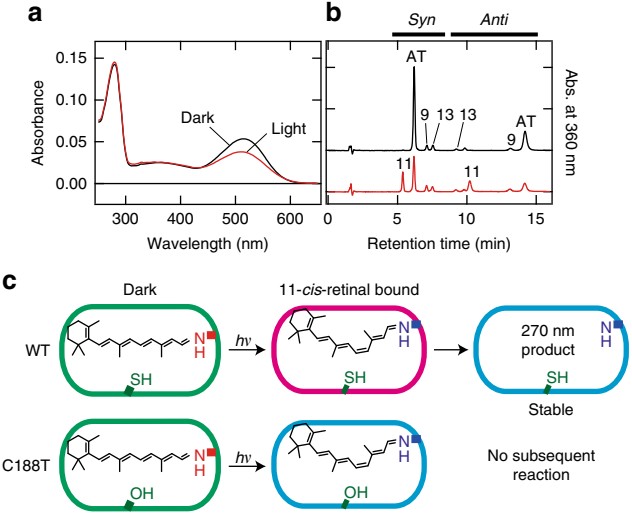

**Fig. 3** Amino acid residue responsible for the 270-nm product. **a** Absorption spectra of Opn5L1NC C188T mutant before (black) and after (red) >500 nm light irradiation for 2 min at 10 °C. Irradiation caused the formation of a 500-nm product, but the 270-nm product was not formed. **b** HPLC patterns of the retinal oximes extracted from Opn5L1NC C188T mutant before (black) and after (red) >500 nm light irradiation for 2 min. 11-cis-retinal oxime can be extracted from the irradiated pigment. **c** Reaction scheme of WT Opn5L1NC and its C188T mutant. The WT Opn5L1NC is converted to the 500-nm intermediate by trans-to-cis photoisomerization of the chromophore, followed by formation of the stable 270-nm product, while the C188T mutant is photo-converted to an intermediate similar to the 500-nm intermediate of WT, but this intermediate is stable even at 10 °C

Cys188 in Opn5L1NC was substituted with threonine, the corresponding residue in chicken Opn5m, and after solubilization and purification, the C188T mutant of Opn5L1NC showed an absorption maximum similar to that of WT Opn5L1NC (Fig. 3a). Irradiation with >500 nm light caused the formation of a mixture containing the original C188T mutant and its photoproduct. We could subsequently extract 11-cis-retinal chromophore in addition to the original all-trans-retinal chromophore from the mutant sample (Fig. 3b), whereas we could not extract the retinal chromophore from the irradiated WT sample (Fig. 1e). Thus, WT Opn5L1NC is converted to the 500-nm intermediate by trans-to-cis photoisomerization of the chromophore, followed by formation of the stable 270-nm product, while the C188T mutant is photo-converted to an intermediate similar to the 500-nm intermediate of WT, but this intermediate is stable even at room temperature (Fig. 3c). We also verified the function of Cys188 in the mutant Opn5L1 bearing native N- and C- termini (Supplementary Fig. 3e−h).

Next, we tested the formation of the thio-retinal adduct by subjecting Opn5L1NC to protein digestion and liquid chromatography-mass spectrometry (LC-MS). The addition of hydroxylamine to the adduct sample would cause formation of retinal oxime bound to cysteine but not to lysine (Fig. 4a). Thus, a peptide fragment containing cysteine that forms a thio-retinal adduct was expected to be detected by LC-MS after suitable digestion of Opn5L1NC. We prepared the E177K/Q192K mutant of Opn5L1NC to obtain a peptide fragment containing Cys188 by tryptic digestion. The introduction of these

two lysines into the second extracellular loop did not affect the spectral or kinetic properties of the reaction of Opn5L1NC (Supplementary Fig. 8). The sequence of the expected peptide was YGEEPYGTACCIDWK, whose first cysteine would be carbamidomethylated with iodoacetoamide, and whose second cysteine would be linked to retinal (Fig. 4b and Supplementary Fig. 9a). The tryptic digest sample was subjected to HPLC followed by a mass spectrometry. We detected $[M + 2H]^{2+}$ and $[M + 3H]^{3+}$ cations having m/z values corresponding to those of the expected peptides in the samples before and after irradiation in the presence of hydroxylamine (Fig. 4c−e and Supplementary Fig. 9b−d). Light increased the amount of peptide with retinal oxime at the second cysteine and decreased the amount of peptide with carbamidomethylation at that position (Fig. 4f, g, and Supplementary Fig. 9e−j).

In addition, we obtained further evidence for adduct formation using a western blotting-based method (Supplementary Fig. 10). If one assumes that the thio-retinal adduct connects Cys188 and Lys296, the full-length Opn5L1NC should be detectable even after the cleavage of a peptide bond between Cys188 and Lys296 (Supplementary Fig. 10a). To divide Opn5L1NC into two parts, we constructed an Opn5L1NC mutant with a factor Xa recognition site at the third intracellular loop. Additionally, we irradiated an Opn5L1 samples in the presence of a boron-based reducing agent to stabilize the linkage between Cys188 and Lys296 by amination of the Schiff base, because the presence of a reactive protonated Schiff base can result in dissociation of the retinal-thio adduct or hydrolysis of the Schiff base (Supplementary Fig. 10b). Our results showed that, in the presence of a reducing agent that stabilizes the retinal-thio adduct, irradiation with light increased the amount of the full-length Opn5L1NC protein bands even after cleavage within the

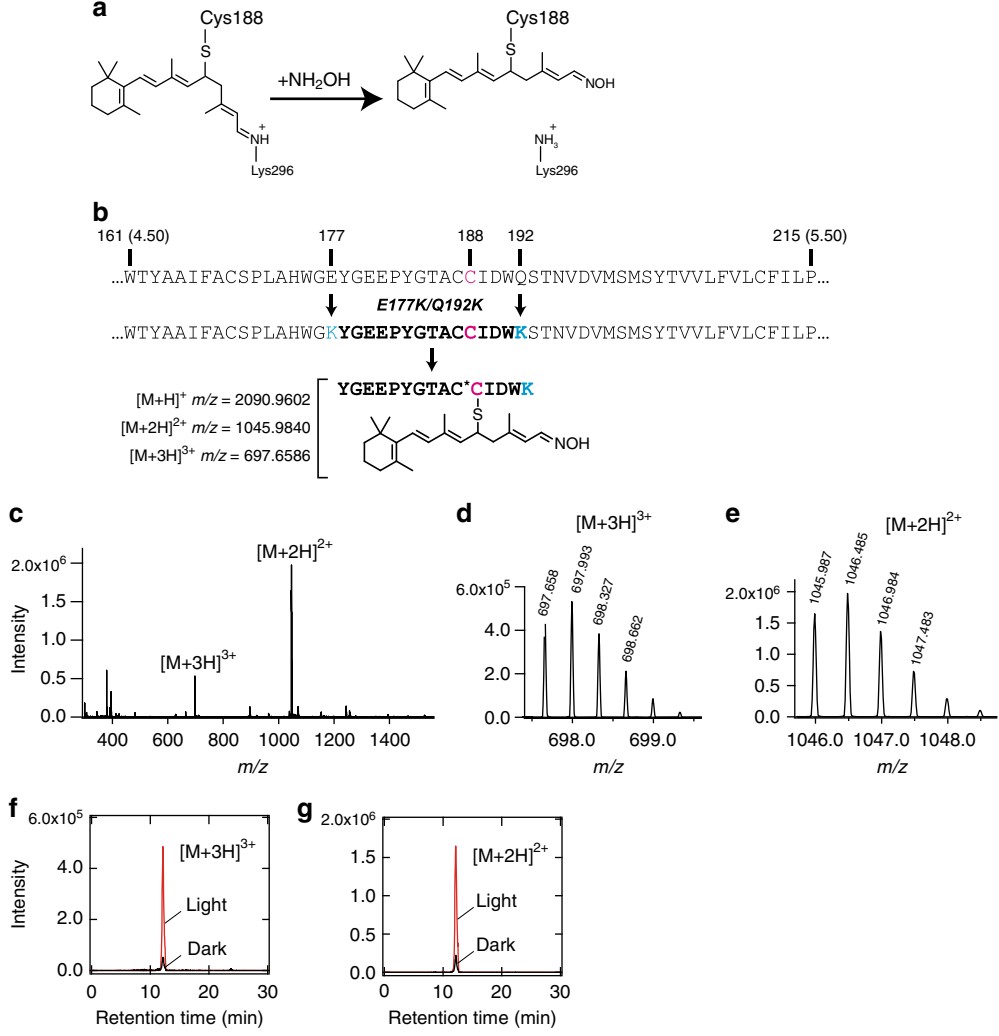

**Fig. 4** Detection of retinal-thio adduct formation in light-irradiated Opn5L1 by LC-MS. **a** Action of hydroxylamine on retinal-thio adduct in Opn5L1. In this figure, we show that $C_{11}$ is the site of adduction of the retinal chromophore to C188. **b** Amino acid sequences of fragments containing C188 in Opn5L1NC and Opn5L1NC E177K/Q192K mutant. Structure of the expected peptide (after light irradiation in the presence of hydroxylamine, breakdown of disulfide bond between C110 and C187, carbamidomethylation of free cysteine, and tryptic digestion) is shown below. Amino acids are numbered based on the bovine rhodopsin numbering system. Numbers shown in parentheses are according to Ballesteros−Weinstein numbering. Asterisk indicates carbamidomethylation. **c** Mass spectrum recorded at the retention time of 12.15 min, peak time of $m/z = 1045.984 \pm 0.05$ corresponding to isotopic $m/z$ of $[M + 2H]^{2+}$ of the peptide YGEEPYGTAC*C(retinaloxime)IDWK. The mass signals of $[M + 2H]^{2+}$ and $[M + 3H]^{3+}$ ions of YGEEPYGTAC*C (retinaloxime)IDWK are indicated. **d**, **e** Enlarged view of the mass signals of $[M + 3H]^{3+}$ (**d**) and $[M + 2H]^{2+}$ (**e**) from **c**, respectively. **f**, **g** LC-MS profiles of $m/z = 697.659 \pm 0.05$ (**f**) and $1045.984 \pm 0.05$ (**g**) corresponding to isotopic $m/z$ of $[M + 3H]^{3+}$ and $[M + 2H]^{2+}$ of the peptide YGEEPYGTAC*C (retinaloxime)IDWK, respectively. The traces obtained from the light-irradiated (red) and the dark-adapted (black) samples are shown

third intracellular loop (Supplementary Fig. 10e). These results further support the molecular model of light-dependent formation of a thio-retinal adduct with Cys188. Cys188 is completely conserved in the Opn5L1 subgroup, which supports the notion that the formation of a thio-retinal adduct is shared among the members of the Opn5L1 subgroup (Supplementary Fig. 11).

**Photosensitivity and tissue localization of Opn5L1**. To obtain insight into the physiological roles of Opn5L1, we first investigated the photosensitivity of Opn5L1NC. As Opn5L1NC is completely "bleached" upon absorption of visible light at 0 °C, we successively irradiated Opn5L1NC with 500-nm-light and recorded the absorption spectrum upon each irradiation. Then, we plotted the relative peak absorbance of Opn5L1NC against irradiation time on a semi-logarithmic scale and estimated the photosensitivity by linear regression of these plots

(Supplementary Fig. 12a−c). The photosensitivity of Opn5L1 thus obtained was 0.73 relative to that of bovine rhodopsin, which is comparable to the photosensitivities of other opsins such as cone pigments and melanopsin[26,27]. Using this value together with the molar extinction coefficient (44,700 at 510 nm, Supplementary Fig.12d, e), the quantum yield of Opn5L1 was calculated to be 0.42, which is equivalent to that of the photoreaction from acid metarhodopsin to the dark state in octopus rhodopsin[28]. Thus, the photochemical parameters of Opn5L1 are quite similar to those of other opsins that function as an optical switch.

We next investigated the tissue distribution of Opn5L1 by performing in situ hybridization. Hybridization signals of *Opn5L1* were detected in multiple regions in chicken brain and retina (Fig. 5). In chicken telencephalon, hybridization signals of *Opn5L1* were broadly distributed in the mesopallium, but not in the hyperpallium or nidopallium (Fig. 5b−e, Supplementary Fig. 13a, b). In addition, we found hybridization signals in the

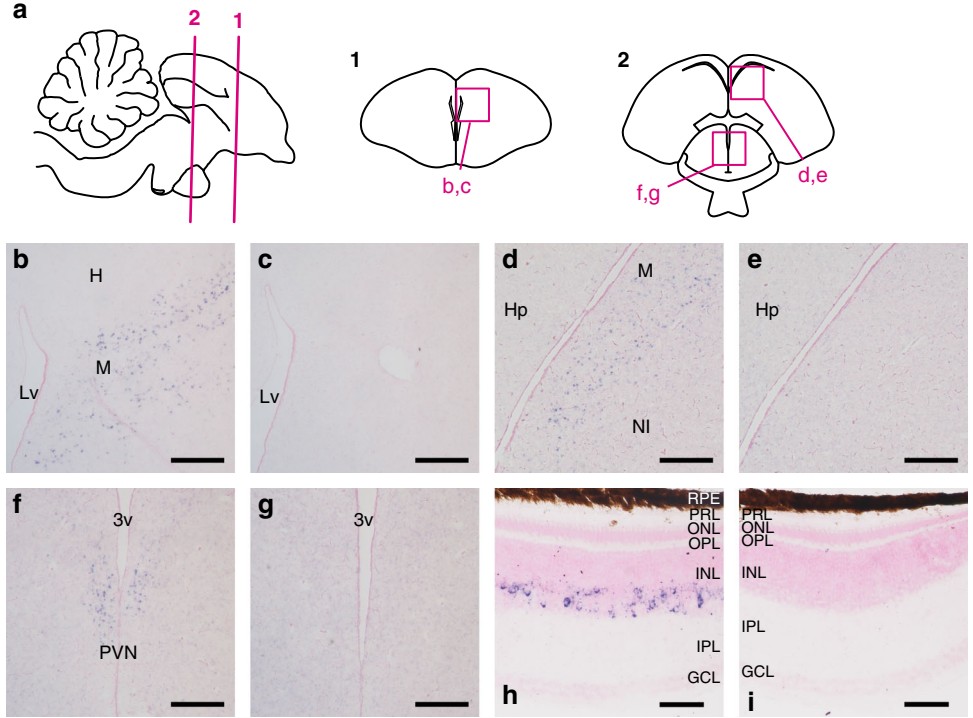

**Fig. 5** Distribution of *Opn5L1* mRNA in chicken brain and retina. **a** Schematic drawings of chicken brain to show the approximate positions of frontal sections. Numbered lines in sagittal drawing indicate the positions of frontal sections numbered 1 and 2. Magenta boxes show the areas of panels **b**−**g**. **b**−**g** *Opn5L1* mRNA in the chicken brain. Sections were hybridized with *Opn5L1* antisense (**b**, **d**, **f**) and sense (**c**, **e**, **g**) probes. Panels (**c**), (**e**), and (**g**) show the tissue sections consecutive to (**b**), (**d**), and (**f**), respectively. **h**, **i** *Opn5L1* mRNA in the chicken retina. Sections were hybridized with *Opn5L1* antisense (**h**) and sense (**i**) probes. All the sections were counterstained with Nuclear Fast Red. Scale bars: **b**−**g** 300 μm; **h**, **i** 50 μm. Lv lateral ventricle, H hyperpallium, M mesopallium, Hp hippocampus, NI intermediate nidopallium, 3v third ventricle, PVN paraventricular nucleus, RPE retinal pigment epithelium, PRL photoreceptor layer, ONL outer nuclear layer, OPL outer plexiform layer, INL inner nuclear layer, IPL inner plexiform layer, GCL ganglion cell layer

**Fig. 6** Chromophore structural changes in the photocycle of Opn5L1. Opn5L1 has all-*trans*-retinal as a chromophore, near which cysteine residue at position 188 (Cys188) is situated (top left). Light causes all-*trans*-to-11-*cis* isomerization of the chromophore (top right). The isomerization is followed by adduct formation between the chromophore and the thiol group of Cys188, resulting in conversion of C11=C12 double bond to a single bond in the chromophore (bottom right). As the C11−C12 single bond is thermally rotated, adduct dissociation occurs when the C11−C12 bond is in a *trans* conformation (bottom left). Subsequent reformation of the all-*trans*-retinal chromophore recovers the original state of Opn5L1 (top left)

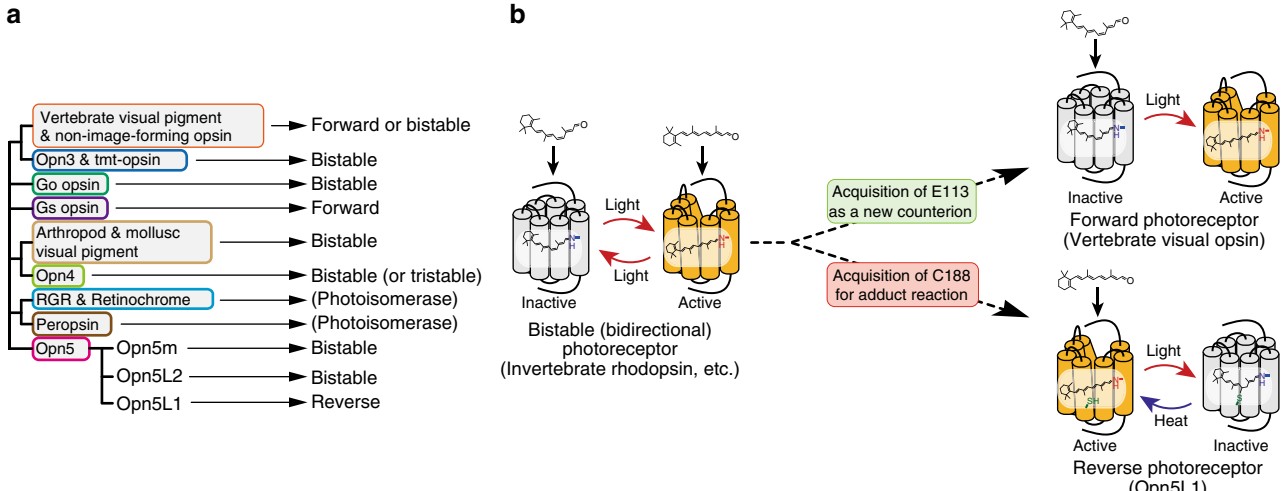

**Fig. 7** A possible scenario of evolution of forward and reverse photoreceptors from an ancestral bistable opsin. **a** Phylogenetic view of opsin family and photochemical properties of the groups in the family. As many members in most of the groups show the bistable nature, such bistability should be an ancestral type of the photochemistry of opsins. **b** Schematic view of possible evolutionary branching into forward and reverse photoreceptors. Loss of ability to directly bind all-*trans*-retinal and undergo its *trans-cis* photoisomerization produced forward photoreceptors. On the other hand, reverse photoreceptors lost the ability to bind 11-*cis*-retinal along with the ability to undergo the photo-conversion reaction from the inactive state to active state

paraventricular nucleus (PVN) in the chicken diencephalon (Fig. 5f, g, Supplementary Fig. 13c). Furthermore, hybridization signals were observed in the inner side of the inner nuclear layer in chicken retina, which indicates that a subpopulation of amacrine cells express *Opn5L1* (Fig. 5h, i, Supplementary Fig. 13d).

## Discussion

Based on the above results, we summarized the photochemical and subsequent thermal reactions of Opn5L1 in Fig. 6. Light causes a *trans*-to-*cis* isomerization of the retinal chromophore of Opn5L1, resulting in formation of the 500-nm intermediate having an 11-*cis*-retinal chromophore. As a conjugated system connected to iminium can act as a nucleophile acceptor, the 11-*cis*-retinal chromophore in the intermediate reacts with the thiol group of nearby Cys188 through nucleophilic addition, resulting in formation of the 270-nm product. In this product, the $C_{11}=C_{12}$ double bond of the retinal chromophore would be converted to a single bond by adduct formation, allowing free rotation about the resulting $C_{11}-C_{12}$ bond. Adduct dissociation would occur with a certain probability when the $C_{11}-C_{12}$ bond is in a *trans* conformation, and reformation of the $C_{11}=C_{12}$ double bond results in reformation of the original Opn5L1 with the all-*trans*-retinal chromophore. Several reports have described the reaction of 1,4- or 1,6-addition of thiol to a conjugated double bond system having iminium or amine at its end[29–33]. It is well known that light oxygen voltage proteins, which are widespread in plants and microbes, form an adduct between their chromophore, flavin, and an adjacent cysteine residue upon light absorption. In such cases, the adduct is formed in the excited state or its subsequent triplet state with a time constant of nanoseconds to microseconds[34,35]. In contrast, a retinal-cysteine adduct was formed in the ground-state intermediate having an 11-*cis*-retinal chromophore, which was formed by light irradiation of Opn5L1. Thus, the formation of the retinal-cysteine adduct in Opn5L1 is a thermal reaction, although the detailed mechanism of the adduct formation is not yet clear. In addition, we did not directly confirm that $C_{11}$ is the acceptor for conjugate addition of thiol in this study, although this would be the most straightforward explanation. Further investigations employing various techniques, including UV-

resonance Raman spectroscopy, NMR spectroscopy, and X-ray diffraction, would be necessary to achieve a complete understanding of the reaction mechanism.

We have demonstrated that Opn5L1 is a retinal receptor that can function as a reverse photoreceptor. The molecular properties of Opn5L1 are quite different from those of vertebrate visual rhodopsin. However, from an evolutionary point of view, both of these opsins have diverged into unidirectional photoreceptors (reverse and forward, respectively) from bistable (bidirectional) opsins, which exhibit bidirectional photoreactions, and are widely distributed among various opsin groups (Fig. 7).

Many members of various opsin subgroups, including invertebrate rhodopsins, are bistable (bidirectional) opsins that form stable resting (inactive) and active states through the direct binding of 11-*cis*- and all-*trans*-retinals, respectively, although they bind preferentially to 11-*cis*-retinal[13,36,37]. These two states are photo-convertible. Hence, bistable opsins function as non-saturable photoreceptors even under continuous illumination by forming a photostationary equilibrium between the active and inactive states[38]. However, one consequence of bistability is that only a fraction of the receptor can be activated, resulting in low efficiency of triggering of the transduction of light signals. Vertebrate rhodopsins are thought to have diversified from an ancestral bistable opsin by displacing a counterion for stabilizing the protonated Schiff base from the ancestral position Glu181 to the newly acquired Glu113[10,39,40]. Vertebrate rhodopsins bind 11-*cis*-retinal exclusively to form a stable inactive state which is photo-converted to a metastable UV light-absorbing state that is able to activate G proteins efficiently[10]. This active state (metarhodopsin II) cannot revert to the inactive dark state[41]. Therefore, vertebrate rhodopsins are optimized forward photoreceptors that can transduce the light signal precisely with a higher signal-to-noise ratio than that of bistable opsins, as long as sufficient 11-*cis*-retinal is available[42].

Opn5L1 is also expected to have evolved from bistable opsins. However, Opn5L1 binds all-*trans*-retinal exclusively and is inactivated by light. Additionally, through the acquisition of a cysteine residue at position 188, it can form the covalently stabilized resting inactive state in a light-dependent manner. This resting state has no absorbance in the visible or near-UV region and cannot photo-revert to the active state. Therefore, Opn5L1 is

an optimized reverse photoreceptor that can convert fully to the inactive state, which is the opposite of the evolutionary direction of vertebrate rhodopsins that can convert fully to the active state. The acquisition of the cysteine residue in Opn5L1 also led to the ability to undergo slow thermal conversion of the inverse agonist to the agonist by utilizing retinal as an acceptor for nucleophilic addition. This is in contrast to the classical paradigm stating that light energy is necessary to convert the inverse agonist to the agonist in vertebrate rhodopsins and bistable opsins. Such thermal regeneration of the active state enables Opn5L1 to be inactivated and activated repeatedly by one retinal in the absence of a retinal supply system.

Opn5L1 rapidly loses its Gi activation ability on absorption of a photon and then gradually recovers this ability in the dark. This molecular property could be utilized as a molecular timer to measure the time elapsed since the last illumination, e.g., after sunset. To obtain insight into Opn5L1's physiological function, we investigated the localization of Opn5L1 in chicken brain and retina. The results showed that Opn5L1 is expressed in the mesopallium and PVN in the chicken brain. It was reported based on experiments using birds that light can penetrate to the relatively superficial mesopallium and even the hypothalamus[43], suggesting that Opn5L1 works as a reverse photoreceptor even in the hypothalamus. Based on the intensity of the solar light reaching the base of brain ($0.2–2.0 \times 10^{12}$ photons $m^{-2} s^{-1} nm^{-1}$ (370–600 nm))[43] and the size of cells in PVN (10 μm radius)[44], one can estimate the rate of photoresponse of Opn5L1 in one neuron to be one per several minutes, assuming that the density of Opn5L1 is equivalent to that of melanopsin in retinal ganglion cells ($3 \mu m^{-2}$)[45]. If the density of Opn5L1 is equivalent to that of rhodopsin in the disk membranes of rod photoreceptor cells ($25{,}000 \mu m^{-2}$)[46], the estimate rise to about 20 molecules of Opn5L1 in 1 s. Although there have been no reports on the expression of opsins in the avian mesopallium or on physiological functions that would imply the occurrence there of molecular functions that could be attributed to Opn5L1 in the avian hypothalamus, Opn5L1 might participate in the photoreceptive control of complex functions and behaviors in cooperation with forward photoreceptor and/or bistable opsins. Opn5L1 is also expressed in a subset of amacrine cells in the retina, suggesting that Opn5L1 is responsible for the regulation of visual processing mediated by the amacrine cells.

In conclusion, bistable opsins evolutionarily acquired specific amino acids at appropriate positions in the protein moieties to have diverged into two opposing unidirectional photoreceptors with an increased output signal-to-noise ratio during signal transduction, one of which has acquired the ability to spontaneously regenerate its activity in the dark.

## Methods

**Recombinant protein preparation**. The cDNA of chicken Opn5L1 (NCBI accession number AB368181) was modified by replacement of its N- and/or C-termini sequences by those of *Xenopus tropicalis* Opn5m (XM_002935990) to improve the expression level (Supplementary Fig. 2). The modified Opn5L1 was tagged with the epitope sequence of the anti-bovine rhodopsin monoclonal antibody Rho1D4 (ETSQVAPA) at the C terminus and was inserted into the mammalian expression vector pCAGGS. The plasmid DNA was transfected into HEK293T cells[47] using the calcium phosphate method. In the assessment of the preference for retinal isomers and the G protein activation assay by membrane fractions, retinal was not added to cells in culture. In the other experiments, the medium was supplemented with 5 μM all-*trans*-retinal 24 h after transfection and the cells were kept in the dark thereafter to prepare all-*trans*-retinal-bound pigment. Forty-eight hours after the transfection, the HEK293T cells were collected. For the assessment of the preference of retinal isomer, collected cells were incubated with 10 μM 11-*cis*-retinal or all-*trans*-retinal at 4 °C for 8 h in the dark. The following procedures were carried out on ice under dim red light unless otherwise noted. The proteins were extracted with 1% dodecyl maltoside (DM) in buffer A (50 mM HEPES (pH 7.0), 140 mM NaCl and 3 mM $MgCl_2$). To obtain purified Opn5L1, the DM-extract was applied to Rho1D4 antibody-conjugated Sepharose.

The column-bound, purified protein was eluted with buffer A (Fig. 1a, Supplementary Figs. 3 and 11) or buffer B (50 mM HEPES (pH 7.0), 20 mM NaCl and 3 mM $MgCl_2$) (Figs. 1d, f, h, 3a, Supplementary Figs. 8 and 10) supplemented with 0.02% DM and 0.45 mg mL$^{-1}$ synthetic peptide of the Rho1D4 epitope sequence. The concentration of NaCl in all the samples was adjusted to 140 mM after elution. The membrane fractions of HEK293T cells were prepared by the standard sucrose flotation method[48].

**Spectrophotometry**. Absorption spectra of the samples were recorded using a conventional spectrophotometer (Shimadzu UV2450). To regulate the sample temperature, an optical cell-holder was connected to a Neslab RTE-7 temperature controller. The temperature of the sample used for spectral recording is shown in the legend of each figure. The sample was irradiated with light from a 1-kW tungsten halogen lamp (Rikagaku Seiki) that had been passed through a glass cut-off filter (VY52; Toshiba Co., Ltd.). For measurement of time-resolved absorption spectra on a sub-second timescale, a customized CCD spectrophotometer (C10000 system, Hamamatsu Photonics Co., Ltd.) was used (Fig. 1f and Supplementary Fig. 8b). By using this spectrophotometer, absorption spectra (257–728 nm at every 0.23 nm) were continuously recorded at time intervals of 200 μs. The sample temperature was kept at 37 ± 0.1 °C by using a cell holder equipped with a Peltier device. The sample was irradiated with light passing through a glass cut-off filter VY52 from a short-arc xenon flash lamp (~170 μs; SA-200F, Nissin Electronic Co., Ltd.) 100 ms after starting the measurement. In Fig. 1f and Supplementary Fig. 8b, unreacted component was subtracted from each measured spectrum after irradiation.

**Retinal extraction and HPLC analysis**. Retinal isomeric composition was analyzed by HPLC (Shimadzu LC-10AT VP) with a silica column (150 × 6.0 mm, A-012–3; YMC) and a solvent composed of 98.8% (v/v) benzene, 1.0% (v/v) diethyl ether, and 0.2% (v/v) 2-propanol[26]. The retinal chromophores of purified samples were extracted in the form of retinaloximes as follows. Immediately after samples were light-exposed, or kept in dark, a 150 μL aliquot of sample was treated with 15 μL of 1 M hydroxylamine and 150 μL of methanol. Retinaloximes were then isolated by phase separation, with 150 μL of dichloromethane and 1 mL of hexane. After mixing and centrifugation, the upper organic layer was collected, dried by addition of small amount of solid anhydrous sodium sulfate, and evaporated by spraying $N_2$ gas. Dried samples were dissolved in 20 μL of hexane and analyzed by HPLC. Extraction of retinal from the early photo-intermediate of Opn5L1 was performed as follows[25]. The purified Opn5L1 was supplemented with 67% (v/v) glycerol and cooled in an ethanol/dry ice bath. After irradiation with light passing through VY52 for 1 min, the sample was mixed with two volumes of cooled methanol containing 100 mM hydroxylamine and incubated for 30 min in the cooling bath. The sample was slowly warmed up to 0 °C with vigorous mixing, and submitted to the retinal extraction and analysis protocol described above. In parallel, retinal was extracted from a sample kept on ice for 30 min after light irradiation in the ethanol/dry ice bath.

**Liquid chromatography-mass spectrometric analysis**. To identify the formation of the intramolecular adduct in Opn5L1, tryptic digestion and LC-MS analysis were performed as follows. Purified Opn5L1NC E177K/Q192K mutant was supplemented with 100 mM neutralized hydroxylamine and separated into two 150 μL aliquots. One was irradiated with light passing through a glass cut-off filter VY52 for 5 min, and the other was kept on ice in the dark. Each aliquot was denatured and proteins were precipitated by addition of 0.9 mL of chloroform/methanol/water (1:3:2) solution. The precipitates were then solubilized with 5 μL of 2% RapiGest (Waters), followed by addition of 2.2 μL of DTT solution (55 mM) and incubation at 55 °C for 30 min to cleave disulfide bonds in the protein. To alkylate free cysteine formed in the protein, iodoacetamide solution (120 mM) was added to the sample and the sample was incubated at room temperature for 30 min in the dark. The DTT solution (2.2 μL) was added to the sample to quench residual iodoacetamide in the sample. After that, trypsin (Promega) solution (3.2 μL) was added to the sample and the mixture was incubated at 37 °C for 18 h to digest the protein. The ratio of trypsin to Opn5L1 was 1:5 weight:weight. The sample was then acidified by addition of formic acid, and desalted using MonoSpin C18 (GL Sciences). The peptide mixture was separated on an Xbridge BEH C18 column (Waters) using a linear gradient from 5 to 100% of acetonitrile containing 0.1% formic acid applied over a period of 25 min and a subsequent 5-min isocratic period at 100% acetonitrile containing 0.1% formic acid at a flow rate of 0.2 mL min$^{-1}$. The effluent peptides from the HPLC were directly infused into an electrospray ion source of an LTQ-Orbitrap Discovery mass spectrometer (Thermo-Fisher Scientific). MS and MS$^n$ spectra were obtained using FTMS and ion trap analyzers, respectively. All of the ions were detected in positive mode. LC-MS data were analyzed using mass++ software[49].

**Factor Xa cleavage and western blotting analysis**. To confirm the formation of the intramolecular adduct in Opn5L1, we prepared Opn5L1NC and C188T mutant constructs that contained a factor Xa recognition site and a FLAG tag sequence. Hereafter we refer to the protein expressed from the Opn5L1NC construct as Opn5L1_XaFLAGicl3. Experimental procedures for factor Xa cleavage and western blotting were as follows. The purified Opn5L1_XaFLAGicl3 (20 μL) was irradiated

with light passed through a glass cut-off filter VY52 at room temperature for 1 h in the presence or absence of 8 mM 2-picoline borane. The samples were then incubated at 37 °C for 2 h in the presence or absence of factor Xa (final concentration: 50 µg mL$^{-1}$). The samples after incubation were mixed with SDS sample buffer containing DTT (final concentration: 200 mM). Finally, aliquots of the samples were subjected to SDS-PAGE. The protein fragments separated by SDS-PAGE were transferred to a polyvinylidene difluoride membrane, and analyzed by standard western blotting procedures using anti-FLAG M1 (10 µg mL$^{-1}$; Sigma F3040) and Rho1D4 (10 µg ml$^{-1}$)[50] antibodies.

**G protein activation assay.** Activation of G protein by Opn5L1 was monitored by the radionucleotide filter binding assay, which measures GDP/GTPγS exchange by G protein[13]. It should be noted that Opn5L1N, Opn5L1 bearing the native C-terminus (Supplementary Fig. 2) was used instead of Opn5L1NC for measuring G protein activation efficiency (Fig. 1c, j and Supplementary Fig. 4). The assay was performed by using purified Opn5L1N (Fig. 1c) or a subcellular membrane fraction containing Opn5L1N (Fig. 1j and Supplementary Fig. 4) at 25 °C. The membrane fraction was mixed with all-*trans*-retinoid and incubated for at least 1 h before the assay (Supplementary Fig. 4). For measurement of the recovery of G protein activation efficiency (Fig. 1j), Opn5L1 in membrane fractions was irradiated with light passed through a glass cut-off filter VY52 for 2 min on ice, and then incubated at 37 °C for 0, 0.5, 1, 2, 4, or 8 h. After that, G protein activation efficiencies of the incubated samples were estimated with or without additional light irradiation for 2 min. The assay mixture contained 0.01 (Fig. 1c) or 0.015 (Fig. 1j and Supplementary Fig. 4) % DM, 1 µM [$^{35}$S]GTPγS, 140 mM NaCl, 8 mM MgCl$_2$, 1 mM DTT, 50 mM HEPES (pH 7.0), 4 µM GDP, 110 (Fig. 1c), 150 (Fig. 1j), or 32 (Supplementary Fig. 4) nM Opn5L1N, and 0.6 µM Gi. After a certain time period from the start of the GDP/GTPγS exchange reaction by addition of [$^{35}$S]GTPγS, an aliquot of the assay mixture was transferred into 200 µL of stop solution (20 mM Tris-HCl, 100 mM NaCl, 25 mM MgCl$_2$, 1 µM GTPγS, 2 µM GDP, pH7.4). [$^{35}$S] GTPγS bound to G proteins was trapped by passing the solution through a nitrocellulose membrane and quantified with a liquid scintillation counter (Tri-Carb 2910 TR; PerkinElmer).

**Photosensitivity, extinction coefficient and quantum yield.** Photosensitivity of Opn5L1NC and bovine rhodopsin was determined by UV-visible absorption spectroscopy[26]. Recombinant protein of bovine rhodopsin was prepared similarly to Opn5L1NC, with the exception that bovine rhodopsin was regenerated with 11-*cis*-retinal. Bovine rhodopsin was supplemented with 50 mM hydroxylamine for the measurement of the photosensitivity. The samples were irradiated successively with a light passed through an interference filter (peak transmittance, 500 nm; half-band width, 5 nm) and neutral-density filters. Absorption spectra were measured each time before and after irradiation. Finally, the samples were completely bleached by exposure for a sufficient time to light passed through a glass cut-off filter VY52. The relative amount of residual pigment after each irradiation was plotted against the duration of light irradiation on a semi-logarithmic scale. Using linear regression, the photosensitivity of Opn5L1NC was calculated relative to that of bovine rhodopsin[51].

The molar extinction coefficient of Opn5L1NC was determined by the acid denaturation method[39]. After recording the spectrum of the dark state, hydrochloric acid was added to the sample (final pH < 1.5). Acidification denatured the pigment and trapped the chromophore as a protonated all-*trans*-retinylidene Schiff base with an absorption maximum at 440 nm. As a standard, bovine rhodopsin was also denatured by acidification (final pH 1.7) after irradiation with light passing through a glass cut-off filter (O55; Toshiba Co., Ltd.) for 6 min to produce an analogous chromophore state. The molar extinction coefficient of Opn5L1NC was estimated based on that of rhodopsin (40,600 M$^{-1}$ cm$^{-1}$ at 500 nm)[52].

As photosensitivity is proportional to the product of quantum yield and molar extinction coefficient[51], the quantum yield of Opn5L1NC could be estimated relative to that of bovine rhodopsin (0.65)[53] based on the separately determined values of photosensitivity relative to that of bovine rhodopsin and molar extinction coefficient.

**Animals and in situ hybridization.** Animals in these experiments were handled under the guidelines established by the Ministry of Education, Culture, Sports, Science and Technology of Japan, and with approval by the Animal Care and Use Committee of Kyoto University (permit number: H2523, H2622, H2718, and H2815). Nine-day-old male chicks were purchased from a local hatchery (Sato Hatchery, Ltd., Kyoto, Japan). They were killed by intraperitoneal injection of 100 mg kg$^{-1}$ pentobarbital sodium immediately after transport. Eyes and brains were dissected after cardiac perfusion fixation with PBS-buffered 4% paraformaldehyde and were post-fixed in the same solution. Fixed tissues were frozen in a deep freezer at −80 °C in OCT compound (tissue tech) after cryoprotection with PBS-buffered 20% sucrose. Sixteen micrometer sections were cut from frozen tissues and were attached to glass slides (MAS-GP typeA coated glass slide, Matsunami Glass Co., Ltd.). Specimens were stored in a dry chamber at −20 °C.

The preparation of RNA probes and in situ hybridization were performed as follows[54]. For synthesis of *Opn5L1* RNA probe, we first performed 5′ and 3′ rapid amplification of cDNA ends to acquire the sequences of 5′ and 3′ untranslated regions of chicken *Opn5L1* mRNA using primers listed in Supplementary Table 1. Based on the obtained sequences, *Opn5L1* cDNA 2999 bp in length (accession number LC360414) was amplified from chicken brain whole cDNA (ZYAGEN) by using primers: 5′-ACAAGCTGCACGGCCACAACCGC-3′ (forward) and 5′-CAAAGAGCAAACACAGCACAAAAAATGTCC-3′ (reverse). Digoxigenin-labeled RNA probes were synthesized from the *Opn5L1* cDNA inserted into pBluescript KS(+). All the following procedures were carried out at room temperature unless otherwise noted. Specimens were successively bathed in PBS-buffered 4% PFA for 15 min, methanol for 30 min, PBS for 5 min, Tris buffer (50 mM Tris-HCl, 5 mM EDTA, pH 7.6) containing 2 µg mL$^{-1}$ (eye) or 10 µg mL$^{-1}$ (brain) proteinase K at room temperature (eye) or 37 °C (brain) for 15 min, PBS for 5 min, and hybridization buffer (0.75 M NaCl, 75 mM sodium citrate, 0.2 mg mL$^{-1}$ yeast tRNA, 0.1 mg mL$^{-1}$ heparin sodium, 1× Denhardt's solution, 0.1% (v/v) Tween, 0.1% (w/v) CHAPS, 5 mM EDTA, 50% (v/v) formamide) at 65 °C for 3 h. After that, digoxigenin-labeled RNA sense or antisense probe diluted with hybridization buffer (final concentration: 0.17 µg mL$^{-1}$) were applied to the tissue sections and were incubated for about 40 h at 65 °C. After hybridization, they were successively rinsed in SSC buffer (0.15 M NaCl, 15 mM sodium citrate, pH 7.0) containing 50% formamide for 15 min and for 1 h at 65 °C, one-fifth diluted SSC buffer for 1 h at 65 °C, and MABT (100 mM maleate, NaCl, 0.1% Tween 20, pH 7.5) for three times 30 min. After rinsing, the sections were incubated with blocking buffer (1% (v/v) BSA, 10% (v/v) sheep normal serum and 0.08% (v/v) Triton-X100 in PBS) for 30 min and were incubated with anti-digoxigenin Fab fragment conjugated with alkaline phosphatase (1:2000 dilute; Roche 11093274910) overnight at 4 °C. The slides were subsequently washed three times with MABT for 30 min, and twice with AP reaction buffer (100 mM Tris-HCl, 50 mM MgCl$_2$, 100 mM NaCl, 0.1% Tween 20, pH 9.5). Finally, color development was performed with 50 µg mL$^{-1}$ NBT and 175 µg ml$^{-1}$ BCIP in AP reaction buffer.

**Data availability.** All the raw data of LC-MS analyses have been deposited on ProteomeXchange via the jPOST repository (accession number: PXD008890). Data supporting the findings of this manuscript are available from the corresponding author upon reasonable request.

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

## Acknowledgements

We thank Prof. R.S. Molday for the generous gift of a Rho1D4-producing hybridoma, Prof. S. Koike for HEK293T cell line, Prof. H. Niwa for pCAGGS vector, and The Institute for Amphibian Biology (Hiroshima University, Hiroshima, Japan) for *X. tropicalis* through the National Bio-Resource Project of the Ministry of Education, Culture, Sports, Science and Technology of Japan (MEXT). We are also grateful to Prof. S. Noji and Ms. S. Fujita-Yanagibayashi for fruitful discussions, and Prof. B. Leonid, Prof. M. Kono, Dr. E. Nakajima, and Dr. T. Matsuyama for critical reading of our manuscript and invaluable comments. This work was supported by Grants-in-Aid for Scientific Research 25251036 and 16H02515 (to Y.S.) and 24115509 and 15H00812 (to T.Y.) from the Japanese Ministry of Education, Culture, Sports, Science and Technology, a grant from the Takeda Science Foundation (to T.Y.) and from Daiichi Sankyo Foundation of Life Science (to T.Y.) and CREST, JST JPMJCR1753 (to T.Y.)

## Author contributions

K. Sato, T.Y., H.O., and Y.S. designed research. K. Sato, T.Y., A.T., H.G., M.M., K. Sakai, and A.W. performed research. Y.M., S.T., and Y.I. contributed new reagents and analytic tools. K. Sato, T.Y., H.O., A.T., H.G., K.O., M.M., Y.M., K. Sakai, A.W., and Y.S. analyzed data. K. Sato, T.Y., and Y.S. wrote the paper.

## Additional information

**Competing interests:** The authors declare no competing interests.

