## [Peer Review File · Nature Communications]

Reviewers' comments:

Reviewer #1 (Remarks to the Author):

This manuscript describes a new type of opsin (Opn5L1) that binds atRAL and activates its G protein (Gi) in the dark. Absorption of a photon isomerizes the atRAL chromophore to 11cRAL, inactivating the opsin. The opsin then thermally self-activates by forming a thio adduct of 11cRAL with Cys188. The adduct decays to yield active opsin coupled to atRAL. This represents a previously undescribed mechanism for opsin activation/deactivation, distinct from the forward bleaching opsins of ciliary photoreceptors and the bistable opsins of rhabdomeric photoreceptors. The manuscript reports significant and novel findings. The study is complete and the results support the authors' conclusions. Although the authors did not confirm adduct formation on C11 of retinaldehyde, this is the most likely site and the conclusions would be the same if adduct formation occurred on another carbon.

Minor points to address before publication:

Not all opsins are GPCR's, as stated in Abstract and Introduction. RGR-opsin and retinochrome are likely photoisomerases that do not interact with a G protein. The authors should change the text to say that 'most' opsins are GPCR's.

The name Opn5L1 is inappropriate. The hybrid protein with N- and C-termini from Opn5 should not have the same name as parent Opn5L1 protein. Possible alternative names for the hybrid protein are Opn5L1.5 or Opn5L1.2.

The white sections of the bars in Fig. 2B indicating atRAL are difficult to see. Use another fill color.

Reviewer #2 (Remarks to the Author):

This is a solid, well-written, beautiful experimental work that deserves publication on Nature Comm.

The authors show striking and solid evidences for the existence of a novel vertebrate GPCRs retinal-protein (Opn5L1) that, at odd with all the known GPCR-type opsins that bind to 11-cis retinal in their inactive resting state (and are then photo-converted into their active state by 11-cis to all-trans retinal photoisomerization), binds exclusively to the all-trans retinal (the agonist), thus directly generating an active state. This state may be then deactivated by light through an inverse all-trans to 11-cis photoinduced isomerization that leads to the formation of a covalent adduct (the stable inactive state) between the retinal and a nearby cys residue that is eventually turned back to the reactant (the active state) by a slow thermal re-isomerization.

Thus, Opn5L1 acts as a monostable unidirectional photoreceptor (i.e., exhibiting an exclusive photoreaction), like all the known vertebrate rhodopsins (and in contrast with most GPCR-type opsins that are bistable and bidirectional, i.e. the active-inactive states are interconvertible by light irradiation), but in a REVERSE fashion, i.e. deactivation (instead of activation) occurs via light, while exhibiting a self-regeneration ability through a slow thermal process.

The authors provide compelling evidences for the mechanism discussed above, and for all the different steps involved in the photo-thermal cycle leading to Opn5L1 activation-deactivation. Namely, they show the key role played by a retinal nearby cys residue (188) in forming the covalent adduct and stable inactive state, and they postulate a very plausible mechanism for the thermal back-isomerization.

The N- and C-termini modified Opn5L1 opsin employed here is also shown to bear the very same properties of the wild protein, thus ruling out possible interferences due to those modifications.

Finally, let me fairly stress that I am not an expert in the experimental methods/protocols employed in this study and this report is not intended to disregard major issues that may be possibly raised by experimental experts in this field and that should be taken into high consideration.

That said, I warmly recommend publication of this work on NatureComm. This study eventually provides a new paradigm for activation-deactivation of vertebrate rhodopsins, showing an alternative (reverse!) evolutionary route for the design of unidirectional photoreceptors, a necessary key step for increasing S/N ratio in signal transduction as achieved in vertebrates.

I have only a very minor suggestion. I would suggest to move the following sentence at page 7:

"It should be noted that Opn5L1 bearing native N- and C- termini also showed molecular characteristics indistinguishable from those of Opn5L1 with modified N- and C-termini (Supplementary Fig. 4a-d)."

backward to page 5, right AFTER the following sentence:

"we replaced the N- and C-termini of chicken Opn5L1 with those belonging to other Opn5 proteins with higher yields (Supplementary Fig. 2)."

Reviewer #3 (Remarks to the Author):

The manuscript of Keita Sato belongs to the best manuscripts I have ever reviewed.

It describes a rhodopsin that belongs to a special uncharacterized melanopsin subfamily. The team demonstrated very elegantly that this rhodopsin undergoes photo- and thermal reactions that are very different from all other rhodopsins described so far.

As an invertebrate rhodopsin it binds all-trans-retinal - unusual by itself -, it is active in the dark - also unusual, and it photo-isomerized into 11-cis and becomes inactive. But, most surprisingly the 11-cis photoproducts reacts with the retinal polyene backbone under formation of a thioadduct. This is a spectacular finding because this has been discussed > 40 years ago for animal rhodopsins but turned out to be wrong. However, such thioadduct formation is known for flavin-based photoreceptors and bacterial phytochromes (which I would like to be mentioned because it underpins the great flexibility of nature).

I miss a little the theoretical background in the discussion and an argumentation why this reaction is not unlikely and the authors may reconsider the polarity of excited states. An open question is the nature of thioadduct formation. Is it a thermal reaction or a secondary photochemical reaction similar to LOV-proteins. But, many interesting questions about the mechanism could be discussed that would be by far go beyond the scope of this primary report.

A basic publication in this respect is the one from Mathies and Stryer (PNAS 1976, 73, 2169-73).

The manuscript reflects great rhodopsin knowledge, clean spectroscopy, and unusual analytical skills that are - unfortunately - not found any more very often in these days. The proof of the thioadduct is rock solid (spectroscopy, mutagenesis and Mass spectroscopy), which is the major clue.

I have just one question. Is the protein really active in the dark or could it function as a sensor for retinal ? Could it be that it is an apoprotein most the time and only when the retinal concentration rises rhodopsin is formed and this binding process is the start for a signaling process ?

Reviewer #4 (Remarks to the Author):

This is a well written and comprehensive study of the photochemistry and molecular characteristics of OPN5L1 in vitro. The authors convincingly show that this opsin binds all-trans retinal; that this has the effect of allowing G-protein interaction; and that light isomerizes to 11-cis retinal at the same time switching off the receptor. They also continue to a convincing demonstration that 11-cis retinal isoform is retained by the opsin and thermally relaxes to the active all-trans isoform over hours. These characteristics are unique for opsins and therefore represent a very interesting photochemical description.

The major weakness of this paper is the lack of a functional perspective. Assuming that these characteristics are recapitulated in vivo, what physiological function can the authors conceive for Opn5L? Some speculation is required, but some information is also lacking. Where is Opn5L expressed in the chicken? This could provide some clues as to function.

The authors propose that Opn5L can act as an optical switch, but does this have any physiological relevance in vivo? Can the authors provide an estimate of the rate of Opn5L 'photobleach' in its native environment under real world conditions? A related question that the authors can address is whether applying all-trans retinal can recover 'bleached' Opn5L? If all-trans regenerates Opn5L pigment readily in the dark, then its unlikely that light will have a marked effect on its signaling in vivo and Opn5L could signal retinal concentrations, alternatively if it does not then this indicates light is indeed the physiologically relevant stimulus.

The authors should put the all-trans retinal concentration required to activate Opn5L into context, are these physiological levels?

Specific comments:

Line 98. The retinoid processing machinery of native HEK293S cells (Brueggemann, L. I. & Sullivan, J. M. HEK293S cells have functional retinoid processing machinery. *J. Gen. Physiol.* 119, 593–612 (2002) might be a relevant consideration as a reason as to why all-trans was found in the media after incubation of cells with cis-retinal + OPN5.

Reviewer #5 (Remarks to the Author):

This study reports photochemical and structural properties of Opn5L1, an opsin-type protein whose nominal ligand (i.e., ligand in darkness) is all-trans retinal. The data show that this all-trans retinal is a light-regulated chromophore. Upon illumination, the Opn5L1-bound all-trans retinal undergoes isomerization specifically to 11-cis retinal, and this transition is accompanied by a reduction in Gi-protein activation. This all-trans to 11-cis photoisomerization is followed by a slow reversion, in darkness, of the 11-cis retinal to all trans retinal through a reaction promoted by the thiol group of a specific cysteine in Opn5L1 (cys188).

As noted by the authors, identifying and understanding the nature of new members within the multiple families of opsin-type proteins are topics of major interest and importance to the fields of cell biology and physiology. The manuscript reports highly interesting information on Opn5L1, a member of the Opn5 group with a novel retinal "photocycle" that is remarkably distinct from, e.g., those of the retinal-bleaching visual opsins of vertebrates and the bidirectional (reversibly photoconvertible) visual opsins of invertebrates. Also as noted in the text, the occurrence of Opn5s in multiple non-retinal tissues supports the logic of the notion that Opn5L1 may function in circadian-associated, photoentrainment, or other non-visual light-dependent signaling pathways. Major overall strengths of the study are the logic of the experimental designs, and the thoroughness and presentation clarity of the biochemical analyses. The combined results obtained from spectrophotometry, the characterization of retinal/protein states at differential temperatures, liquid chromatography/mass spectrometry analysis, substitution of a threonine for cys188, and from other peptide preparations/analyses (notably, the experiment demonstrating all-trans retinal's bridging of cys188 to the retinal binding site at lys296) (Supplementary Figure 8) convincingly establish Opn5L1's action on the retinal ligand in light (all-trans to 11-cis) and in darkness (11cis to all-trans).

My only significant criticism concerns the limited information provided on the downregulating action of illuminated Opn5L1, i.e., of all-trans to 11-cis retinal photoisomerization, on Gi-protein activation (Figures 1c, 1j, and Supplementary Figures 3 and 4d). The reported data show that photoisomerization reduces activation of the Gi present. However, the text lacks information that could enable evaluation of how robust is this physiological activity. Because the paper is the first photo-characterization of Opn5L1, and because Opn5L1's potential roles in cell signaling pathways and metabolism remain to be investigated, extensive characterization of the Gi-protein effect – specifically, the magnitude of Opn5L1's action on Gi-protein-targeted downstream biomolecules as a function of the number of Opn5L1s photoisomerized – is probably beyond the scope of the paper. However, a main conclusion drawn by the authors is that Opn5L1 is a physiological "photoreceptor" that in reverse-acting fashion controls some cellular process via Gi downregulation. Thus, even in a first report, greater attention should be given to clarifying how potent is this Gi deactivation signal. First, the authors should analyze their spectrophotometric and related data to derive the quantum efficiency with which a photon absorbed by all-trans Opn5L1 converts the protein to 11-cis Opn5L1. Second, under conditions of their present assays, they should determine how many Gi-proteins are de-activated by a given photoisomerized Opn5L1, and the time course with which this occurs after a brief light flash of calibrated intensity. Third, they could determine how the physiological specific activity of Opn5L1 (i.e., the number of Gi-proteins deactivated per photoisomerized Opn5L1) compares with the physiological specific activity of some other representative retinal-binding opsin (number of G-proteins activated per photoisomerization of, e.g., a visual opsin). Information on these points, although not firmly establishing or ruling out a physiological role of Opn5L1's Gi-protein deactivation, would provide a better grounding of the present findings in the context of photoisomerization efficiency and G-protein regulation by other retinal-binding opsins.

Overall, given the relevance specifically of the 11-cis isomer to physiological functions of retinal-binding opsins, the paper's clear evidence that Opn5L1 illumination specifically generates the 11-cis isomer is immediately intriguing and suggestive of the "photoreceptor" function concluded by the authors. However, it is conceivable that Opn5L1 acts, alternatively, as (merely) a binding protein for all-trans retinal that scavenges or sequesters this widely distributed retinoid, and that Gi-protein regulation by Opn5L1 illumination is a weak side-effect of negligible physiological importance. Further information of the types noted in points 1-3 above could bear on the evaluation of this alternative hypothesis.

Minor points:

What is the effect of reducing retinal's Schiff base linkage to Opn5L's lys296 using borohydride?

What are the binding constants for Opn5L (high-affinity) binding of all-trans and (low-affinity)

binding of 11-cis retinal?

In Figure 1, the semilog plot of the "single exponential functions" in Fig. 1 is appropriate. However, the sigmoid nature of these semilog plots might be confusing to some readers. For clarity, the Figure 1 legend (manuscript lines 562-563) should indicate the full equations for single exponential

functions being used.

Reviewer #1

Comment:

Not all opsins are GPCR's, as stated in Abstract and Introduction. RGR-opsin and retinochrome are likely photoisomerases that do not interact with a G protein. The authors should change the text to say that 'most' opsins are GPCR's.

Response:

We added "most" as you suggested.

Comment:

The name Opn5L1 is inappropriate. The hybrid protein with N- and C-termini from Opn5 should not have the same name as parent Opn5L1 protein. Possible alternative names for the hybrid protein are Opn5L1.5 or Opn5L1.2.

Response:

Thank you for your suggestion. According to your comment, we referred to chicken Opn5L1 having the N-terminus of *Xenopus tropicalis* Opn5m as Opn5L1N and chicken Opn5L1 having both the N- and C-termini of *X. tropicalis* Opn5m as Opn5L1NC in the revised manuscript.

Comment:

The white sections of the bars in Fig. 2B indicating atRAL are difficult to see. Use another fill color.

Response:

According to your comment, we filled the bars with cyan.

Reviewer #2 (Remarks to the Author):

Comment:

I have only a very minor suggestion. I would suggest to move the following sentence at page 7:

"It should be noted that Opn5L1 bearing native N- and C- termini also showed molecular characteristics indistinguishable from those of Opn5L1 with modified N- and C-termini (Supplementary Fig. 4a-d)."

backward to page 5, right AFTER the following sentence:

"we replaced the N- and C-termini of chicken Opn5L1 with those belonging to other Opn5 proteins with higher yields (Supplementary Fig. 2)."

Response:

Thank you for your comment. We moved the text in the "Results" section as you suggested.

Reviewer #3 (Remarks to the Author):

Comment:

I miss a little the theoretical background in the discussion and an argumentation why this reaction is not unlikely and the authors may reconsider the polarity of excited states. An open question is the nature of thioadduct formation. Is it a thermal reaction or a secondary photochemical reaction similar to LOV-proteins. But, many interesting questions about the mechanism could be discussed that would be by far go beyond the scope of this primary report. A basic publication in this respect is the one from Mathies and Stryer (PNAS 1976, 73, 2169-73).

Response:

It was reported that a flavin-cysteine adduct in LOV protein is formed in the excited state or triplet state with a time constant in the range from nanoseconds to microseconds. In contrast, formation of a retinal-cysteine adduct in Opn5L1 occurs with a time constant of 89 milliseconds in the ground-state intermediate having 11-cis-retinal chromophore, which is formed upon irradiation of Opn5L1. Therefore, the formation of retinal-thio adduct in Opn5L1 is a thermal reaction and not a secondary photoreaction. However, the detailed mechanism of thio-adduct formation in Opn5L1 is an open question, as you noted. We added the following sentences to address this point in the revised manuscript:

“It is well known that light oxygen voltage (LOV) proteins, which are widespread in plants and microbes, form an adduct between their chromophore, flavin, and an adjacent cysteine residue upon light absorption. In such cases, the adduct is formed in the excited state or its subsequent triplet state with a time

constant of nanoseconds to microseconds^{34,35}. In contrast, a retinal-cysteine adduct was formed in the ground-state intermediate having an 11-*cis*-retinal chromophore, which was formed by light irradiation of Opn5L1. Thus, the formation of the retinal-cysteine adduct in Opn5L1 is a thermal reaction, although the detailed mechanism of the adduct formation is not yet clear.” (page 13, lines 15–24)

Comment:

I have just one question. Is the protein really active in the dark or could it function as a sensor for retinal ? Could it be that it is an apoprotein most the time and only when the retinal concentration rises rhodopsin is formed and this binding process is the start for a signaling process ?

Response:

Thank you for your valuable comment. From supplementary Fig. 4 showing retinal concentration-dependency of the G protein activation efficiency, EC_{50} for retinal is estimated to be 6.2×10^{-7} M. We found no reports on the endogenous concentration of retinal in chicken tissues, but did find reports on those in mouse tissues ($10^{-6} - 10^{-8}$ M in tissues and 10^{-8} M in serum) (Ziouzenkova et al., *Nat. Med.* 2007; Kane et al., *Anal. Biochem.* 2008). Thus, assuming that retinal concentration in the chicken tissues is similar to that of mouse, Opn5L1 could act as a retinal sensor in the chicken tissues. On the other hand, we estimated the quantum yield (0.42) and molecular extinction coefficient (44,700) in the

revised manuscript, and these values strongly suggest that Opn5L1 acts as a reverse photoreceptor under physiological conditions. Taking this together with the localization of Opn5L1 in the chicken brain and eyes, for which the data have been newly added in Fig. 5, we infer that Opn5L1 could regulate the complex signaling cascades in eyes and brain with other conventional opsins. We added the following sentences addressing this in the revised manuscript:

“EC₅₀ for retinal was estimated to be 6.2×10^{-7} M. Thus, assuming that the retinal concentration in the chicken tissues is similar to that in mouse ($10^{-6} - 10^{-8}$ M)^{22,23}, Opn5L1 could act as a retinal sensor in the chicken tissues.” (page 6, lines 17–20)

Reviewer #4 (Remarks to the Author):

Comment:

The major weakness of this paper is the lack of a functional perspective.

Assuming that these characteristics are recapitulated in vivo, what physiological function can the authors conceive for Opn5L? Some speculation is required, but some information is also lacking. Where is Opn5L expressed in the chicken?

This could provide some clues as to function.

Response:

In response to your comment, we prepared a new figure (Fig. 5) showing the localization of Opn5L1 mRNA in chicken retina and brain. The results showed that Opn5L1 is expressed in a subset of amacrine cells in the retina, suggesting that Opn5L1 plays a role in the regulation of visual processing mediated by the amacrine cells. In addition, Opn5L1 is expressed in the mesopallium of the telencephalon and in the paraventricular nucleus of the diencephalon. It was reported from experiments using birds that light can penetrate to the relatively superficial mesopallium and even hypothalamus, indicating that Opn5L1 might work not only as a retinal sensor but as a reverse photoreceptor even in the hypothalamus. Although there have been no reports on the expression of opsins in the mesopallium or on the photoreceptive functions directly related to the molecular properties of Opn5L1 in the hypothalamus, Opn5L1 could participate in the control of complex functions and behaviors in cooperation with forward photoreceptor opsin and/or bistable opsin. We added the following sentences in the revised manuscript:

“We next investigated the tissue distribution of Opn5L1 by performing *in situ* hybridization. Hybridization signals of *Opn5L1* were detected in multiple regions in chicken brain and retina (Fig. 5). In chicken telencephalon, hybridization signals of *Opn5L1* were broadly distributed in the mesopallium, but not in the hyperpallium or nidopallium (Fig. 5b-e, Supplementary Fig. 13a, b). In addition, we found hybridization signals in the paraventricular nucleus (PVN) in the chicken diencephalon (Fig. 5f, g, Supplementary Fig. 13c). Furthermore, hybridization signals were observed in the inner side of the inner nuclear layer in chicken retina, which indicates that a subpopulation of amacrine cells express *Opn5L1* (Fig. 5h, i, Supplementary Fig. 13d).” (page 12, lines 13–22)

“Opn5L1 rapidly loses its Gi activation ability on absorption of a photon and then gradually recovers this ability in the dark. This molecular property could be utilized as a molecular timer to measure the time elapsed since the last illumination, e.g., after sunset. To obtain insight into Opn5L1’s physiological function, we investigated the localization of Opn5L1 in chicken brain and retina. The results showed that Opn5L1 is expressed in the mesopallium and PVN in the chicken brain. It was reported based on experiments using birds that light can penetrate to the relatively superficial mesopallium and even the hypothalamus⁴³, suggesting that Opn5L1 works as a reverse photoreceptor even in the hypothalamus. Although there have been no reports on the expression of opsins in the avian mesopallium or on physiological functions that would imply the occurrence there of molecular functions that could be attributed to Opn5L1 in the avian hypothalamus, Opn5L1 might participate in the photoreceptive control of complex functions and behaviors in cooperation with

forward photoreceptor and/or bistable opsins. Opn5L1 is also expressed in a subset of amacrine cells in the retina, suggesting that Opn5L1 is responsible for the regulation of visual processing mediated by the amacrine cells.” (page 15, lines 23–24; page 16, lines 1–15)

Comment:

The authors propose that Opn5L can act as an optical switch, but does this have any physiological relevance in vivo? Can the authors provide an estimate of the rate of Opn5L ‘photobleach’ in its native environment under real world conditions? A related question that the authors can address is whether applying all-trans retinal can recover ‘bleached’ Opn5L? If all-trans regenerates Opn5L pigment readily in the dark, then its unlikely that light will have a marked effect on its signaling in vivo and Opn5L could signal retinal concentrations, alternatively if it does not then this indicates light is indeed the physiologically relevant stimulus. The authors should put the all-trans retinal concentration required to activate Opn5L into context, are these physiological levels?

Response:

To clarify the function of Opn5L1 as an optical switch, we investigated the photosensitivity and quantum yield of Opn5L1 and compared them to those of bovine rhodopsin. The photosensitivity of Opn5L1 was 0.73 relative to that of bovine rhodopsin, which is comparable to the values for several opsins such as mouse UV-sensitive cone pigment (0.84 at 359 nm), chicken violet-sensitive

cone pigment (0.83 at 417 nm), and mouse melanopsin (0.65 at 467 nm) (Tsutsui et al., *Biochemistry* 2007; Matsuyama et al., *Biochemistry* 2012). Based on the measurement of the molar extinction coefficient, the quantum yield of Opn5L1 was calculated to be 0.42. This is equivalent to the quantum yield of the photoreaction from acid metarhodopsin to the dark state in octopus rhodopsin (0.43) (Dixon & Cooper *Photochem. Photobiol.* 1987), an example of isomerization of all-trans-retinal to 11-cis-retinal in opsins. Therefore, the photochemical properties of Opn5L1 are quite similar to those of other opsins, which would enable Opn5L1 to function as an optical switch in physiological situations. We added the results in supplementary Fig. 12 and the sentences as follows:

“To obtain insight into the physiological roles of Opn5L1, we first investigated the photosensitivity of Opn5L1NC. As Opn5L1NC is completely “bleached” upon absorption of visible light at 0 °C, we successively irradiated Opn5L1NC with 500 nm-light and recorded the absorption spectrum upon each irradiation. Then, we plotted the relative peak absorbance of Opn5L1NC against irradiation time on a semi-logarithmic scale and estimated the photosensitivity by linear regression of these plots (Supplementary Fig. 12a-c). The photosensitivity of Opn5L1 thus obtained was 0.73 relative to that of bovine rhodopsin, which is comparable to the photosensitivities of other opsins such as cone pigments and melanopsin^{26,27}. Using this value together with the molar extinction coefficient (44,700 at 510 nm, Supplementary Fig.12d, e), the quantum yield of Opn5L1 was calculated to be 0.42, which is equivalent to that of the photoreaction from acid metarhodopsin to the dark state in octopus rhodopsin²⁸. Thus, the photochemical parameters of

Opn5L1 are quite similar to those of other opsins that function as an optical switch.” (page 11, lines 22–24; page 12, lines 1–12)

Next, to check whether or not light-adapted Opn5L1 can be recovered by the addition of all-*trans*-retinal, Opn5L1 in a membrane suspension was incubated at 37°C after light irradiation with or without all-*trans*-retinal. We could not observe an acceleration of the recovery of the dark state (all-*trans*-retinal-bound state) even in the presence of 4.0×10^{-5} M all-*trans*-retinal, about 100-fold higher concentration than EC_{50} (6.2×10^{-7} M). This shows that exogenous all-*trans*-retinal cannot replace the retinal-thio adduct in the bleached Opn5L1. Therefore, light indeed works as a trigger to suppress the activity, accompanied by the subsequent thermal recovery irrespective of exogenous retinal. We added the results in supplementary Fig. 6 and the following sentence:

“It should be noted that the addition of exogenous all-*trans*-retinal did not accelerate the recovery of the 510-nm state (Supplementary Fig. 6), which indicates that the recovery of the original dark state does not occur by the uptake of exogenous all-*trans*-retinal.” (page 8, lines 7–11)

Finally, from supplementary Fig. 4 showing retinal concentration-dependency of the G protein activation, we estimated EC_{50} for retinal to be 6.2×10^{-7} M. We found no reports on the endogenous concentration of retinal in chicken tissues, but did find reports about the concentrations in mouse tissues (10^{-6} – 10^{-8} M in tissues and 10^{-8} M in serum) (Ziouzenkova et al., *Nat. Med.* 2007; Kane et al., *Anal. Biochem.* 2008). Thus, assuming that the retinal concentration in the

chicken tissues is similar to the reported concentrations in mouse, Opn5L1 has EC₅₀ for retinal in the range of the physiological concentration of retinal. We added the following sentences in the revised manuscript:

“EC₅₀ for retinal was estimated to be 6.2×10^{-7} M. Thus, assuming that the retinal concentration in the chicken tissues is similar to that in mouse ($10^{-6} - 10^{-8}$ M)^{22,23}, Opn5L1 could act as a retinal sensor in the chicken tissues.” (page 6, lines 17–20)

Comment:

Line 98. The retinoid processing machinery of native HEK293S cells (Brueggemann, L. I. & Sullivan, J. M. HEK293S cells have functional retinoid processing machinery. J. Gen. Physiol. 119, 593–612 (2002) might be a relevant consideration as a reason as to why all-trans was found in the media after incubation of cells with cis-retinal + OPN5.

Response:

Thank you for your valuable suggestion. According to your comment, we refer to the report in our manuscript as follows:

“This strongly suggests that Opn5L1 does not bind 11-*cis*- but instead binds all-*trans*-retinal, which could be formed through isomerization of 11-*cis*-retinal catalyzed enzymatically by intrinsic retinoid processing machinery in cultured cells¹⁹ or non-enzymatically by lipids²⁰ or nucleophiles²¹ present in cell suspensions during the incubation.” (page 5, lines 24; page 6, lines 1–4)

Reviewer #5 (Remarks to the Author):

Comment:

My only significant criticism concerns the limited information provided on the downregulating action of illuminated Opn5L1, i.e., of all-trans to 11-cis retinal photoisomerization, on Gi-protein activation (Figures 1c, 1j, and Supplementary Figures 3 and 4d). The reported data show that photoisomerization reduces activation of the Gi present. However, the text lacks information that could enable evaluation of how robust is this physiological activity. Because the paper is the first photo-characterization of Opn5L1, and because Opn5L1's potential roles in cell signaling pathways and metabolism remain to be investigated, extensive characterization of the Gi-protein effect – specifically, the magnitude of Opn5L1's action on Gi-protein-targeted downstream biomolecules as a function of the number of Opn5L1s photoisomerized – is probably beyond the scope of the paper. However, a main conclusion drawn by the authors is that Opn5L1 is a physiological “photoreceptor” that in reverse-acting fashion controls some cellular process via Gi downregulation. Thus, even in a first report, greater attention should be given to clarifying how potent is this Gi deactivation signal. First, the authors should analyze their spectrophotometric and related data to derive the quantum efficiency with which a photon absorbed by all-trans Opn5L1 converts the protein to 11-cis Opn5L1.

Response:

Thank you for your valuable comment. We investigated the photosensitivity and quantum yield of Opn5L1 and compared them to those of bovine rhodopsin. The

photosensitivity of Opn5L1 was 0.73 relative to that of bovine rhodopsin, which is comparable to the photosensitivity of several opsins such as mouse UV-sensitive cone pigment (0.84 at 359 nm), chicken violet-sensitive cone pigment (0.83 at 417 nm), and mouse melanopsin (0.65 at 467 nm) (Tsutsui et al., *Biochemistry* 2007; Matsuyama et al., *Biochemistry* 2012). Based on the measurement of molar extinction coefficient, the quantum yield of Opn5L1 was calculated to be 0.42. This is equivalent to that of the photoreaction from acid metarhodopsin to the dark state in octopus rhodopsin (0.43) (Dixon & Cooper *Photochem. Photobiol.* 1987), an example of isomerization of all-*trans*-retinal to 11-*cis*-retinal in opsins. Therefore, the photochemical property of Opn5L1 is quite similar to those of other opsins, which would enable Opn5L1 to function as an optical switch in the physiological situation. We added the results in supplementary figure 12 and the following sentences:

“To obtain insight into the physiological roles of Opn5L1, we first investigated the photosensitivity of Opn5L1NC. As Opn5L1NC is completely “bleached” upon absorption of visible light at 0 °C, we successively irradiated Opn5L1NC with 500 nm-light and recorded the absorption spectrum upon each irradiation. Then, we plotted the relative peak absorbance of Opn5L1NC against irradiation time on a semi-logarithmic scale and estimated the photosensitivity by linear regression of these plots (Supplementary Fig. 12a-c). The photosensitivity of Opn5L1 thus obtained was 0.73 relative to that of bovine rhodopsin, which is comparable to the photosensitivities of other opsins such as cone pigments and melanopsin^{26,27}. Using this value together with the molar extinction coefficient (44,700 at 510 nm, Supplementary Fig.12d, e), the quantum yield of Opn5L1 was calculated to be

0.42, which is equivalent to that of the photoreaction from acid metarhodopsin to the dark state in octopus rhodopsin²⁸. Thus, the photochemical parameters of Opn5L1 are quite similar to those of other opsins that function as an optical switch.” (page 11, lines 22–24; page 12, lines 1–12)

Comment:

Second, under conditions of their present assays, they should determine how many Gi-proteins are de-activated by a given photoisomerized Opn5L1, and the time course with which this occurs after a brief light flash of calibrated intensity.

Response:

In our assay conditions shown in figure 1c, the rate of incorporation of GTPγS into Gi induced by Opn5L1 (0.55 pmol) was 0.34 pmol/min as determined by linear regression. Thus, irradiation on one molecule of Opn5L1 bound to all-trans-retinal deactivates 0.62 molecule of Gi per minute. Additionally, because irradiation of 1.0×10^{20} photons/m⁻² at 500 nm bleaches about half of bovine rhodopsin in solution, the same number of photons at 510 nm would bleach about 40% of Opn5L1 based on the relative photosensitivity. In this irradiation condition, one molecule of Opn5L1 activates 37 molecules of Gi per hour in the dark, and 15 of the 37 are deactivated after a flash irradiation. To assess the activation rate in the physiological condition, we also compared the activation efficiency between Opn5L1 and bovine rhodopsin. The efficiency of Opn5L1 was about 100-fold lower than that of rhodopsin (see details in the

response to next comment). It was reported that in the physiological condition, one molecule of vertebrate rhodopsin can activate from several hundreds to one thousand G protein molecules per second (Leskov et al., *Neuron* 2000; Heck & Hofmann *J. Biol. Chem.* 2001). Taking into account the difference of the activation efficiency, we can estimate that one molecule of Opn5L1 can activate about 10 G protein molecules per second in the same condition. The previous analysis of the turnover rate of G protein activation by diffusible ligand-bound GPCRs showed that one molecule of a GPCR can activate 1 - 25 G protein molecules per second in cells (Vilardaga et al. *Nat. Biotechnol.* 2003), which is roughly equivalent to our estimate for Opn5L1. Therefore, Opn5L1 would behave like other diffusible ligand-bound GPCRs in the cells. We added the following sentences in the revised manuscript, which also incorporate our response to your next comment:

“Furthermore, we compared the G protein activation efficiency of Opn5L1 with those of chicken Opn5m and bovine rhodopsin (Supplementary Fig. 5). The G_i activation efficiency of Opn5L1 was about 5-fold higher than that of chicken Opn5m and 100-fold lower than that of bovine rhodopsin. This efficiency of Opn5L1 is similar to that of bistable opsin, which has a 50-fold lower activation efficiency than bovine rhodopsin¹⁰.” (page 7, lines 7–12)

Comment:

Third, they could determine how the physiological specific activity of Opn5L1 (i.e., the number of G_i -proteins deactivated per photoisomerized Opn5L1)

compares with the physiological specific activity of some other representative retinal-binding opsin (number of G-proteins activated per photoisomerization of, e.g., a visual opsin). Information on these points, although not firmly establishing or ruling out a physiological role of Opn5L1's Gi-protein deactivation, would provide a better grounding of the present findings in the context of photoisomerization efficiency and G-protein regulation by other retinal-binding opsins. Overall, given the relevance specifically of the 11-cis isomer to physiological functions of retinal-binding opsins, the paper's clear evidence that Opn5L1 illumination specifically generates the 11-cis isomer is immediately intriguing and suggestive of the "photoreceptor" function concluded by the authors. However, it is conceivable that Opn5L1 acts, alternatively, as (merely) a binding protein for all-trans retinal that scavenges or sequesters this widely distributed retinoid, and that Gi-protein regulation by Opn5L1 illumination is a weak side-effect of negligible physiological importance. Further information of the types noted in points 1-3 above could bear on the evaluation of this alternative hypothesis.

Response:

According to your comment, we compared the activation efficiency of Opn5L1 with those of visual opsin, bovine rhodopsin and a non-visual opsin, chicken Opn5m (Supplementary Fig. 5). In our assay condition, the Gi activation efficiency of Opn5L1 was about 100-fold lower than that of bovine rhodopsin and 5-fold higher than that of chicken Opn5m. Recent papers about mouse, quail and frog Opn5m showed that Opn5m can detect light signals to contribute to

various physiological functions (Nakane et al., *Curr. Biol.* 2014; Buhr et al., *Proc. Natl. Acad. Sci. USA* 2015; Currie et al., *Proc. Natl. Acad. Sci. USA* 2016). Thus, it has been established that Opn5m is a photosensor. Furthermore, the G protein activation efficiencies of typical bistable opsins are about 50-fold lower than that of bovine rhodopsin (Terakita et al., *Nat. Struct. Mol. Biol.*, 2004). Taking these facts all together, we infer that the G protein activation efficiency of Opn5L1 would be sufficient to induce a physiological response by retinal binding and photoreception. We added the results in Supplementary Fig. 5 and the following sentences:

“Furthermore, we compared the G protein activation efficiency of Opn5L1 with those of chicken Opn5m and bovine rhodopsin (Supplementary Fig. 5). The G_i activation efficiency of Opn5L1 was about 5-fold higher than that of chicken Opn5m and 100-fold lower than that of bovine rhodopsin. This efficiency of Opn5L1 is similar to that of bistable opsin, which has a 50-fold lower activation efficiency than bovine rhodopsin¹⁰.” (page 7, lines 7–12)

Comment:

What is the effect of reducing retinal's Schiff base linkage to Opn5L's lys296 using borohydride?

Response:

In supplementary Fig. 10, we show that we obtained evidence for adduct formation between Cys188 and the retinal by a western blotting-based method.

In the experimental procedures, we needed to block the dissociation of retinal-thio adduct between retinal and Cys188 and the hydrolysis of Schiff base linkage between retinal and Lys296. Reductive conversion of this Schiff base linkage to secondary amine could prevent these reactions and tighten the linkage between Cys188 and Lys296. Sodium borohydride is one of the widely used reducing agents for such reductive amination. However, sodium borohydride also cleaves disulfide bonds. In the procedures, we used factor Xa to cleave the recognition site inserted into intracellular loop3. Because functional factor Xa forms a dimer bridged by a disulfide bond, sodium borohydride could prevent the protease activity of factor Xa. Thus, we instead used 2-picoline borane, which reduces Schiff base linkage selectively but not the disulfide bond in the present study. We revised the sentence as follows to show our intention more clearly:

“Additionally, we irradiated an Opn5L1 samples in the presence of a boron-based reducing agent to stabilize the linkage between Cys188 and Lys296 by amination of the Schiff base, because the presence of a reactive protonated Schiff base can result in dissociation of the retinal-thio adduct or hydrolysis of the Schiff base (Supplementary Fig. 10b).” (page 11, lines 7–11)

Comment:

What are the binding constants for Opn5L (high-affinity) binding of all-trans and (low-affinity) binding of 11-cis retinal?

Response:

As shown in supplementary figure 4, a titration experiment of Opn5L1 with all-*trans*-retinal indicated that the activity significantly increased in the range from 10^{-8} to 10^{-6} M of all-*trans*-retinal, and EC_{50} for retinal was estimated to be 6.2×10^{-7} M. However, as shown in Figure 1, we could not observe the direct binding of 11-*cis*-retinal to Opn5L1, probably because of its much lower affinity compared to that of all-*trans*-retinal. Thus, we could not determine the affinity of 11-*cis*-retinal for Opn5L1. We added the following sentence in the revised manuscript:

“ EC_{50} for retinal was estimated to be 6.2×10^{-7} M.” (page 6, lines 17–18)

Comment:

In Figure 1, the semilog plot of the “single exponential functions” in Fig. 1 is appropriate. However, the sigmoid nature of these semilog plots might be confusing to some readers. For clarity, the Figure 1 legend (manuscript lines 562-563) should indicate the full equations for single exponential functions being used.

Response:

In response to your comment, we revised the legend as follows;

“(g) The absorbance at 495 nm (black circles) and 270 nm (red triangles) plotted against the time after flash irradiation. The time profiles were fitted by single exponential functions $y=y_0-b \times \exp(-t/\tau)$ with the same time constant (solid curves,

$y=0.0059-0.054 \times \exp(t/0.089)$ for 495 nm, and $y=0.167-0.017 \times \exp(t/0.089)$ for 270 nm, $\tau=0.089$ sec). (h) Absorption spectra before (Dark) and 0, 11, 32, 92, 447, and 557 minutes (curves 1-6, respectively) after irradiation with >500 nm light for 2 min at 37°C. (inset) The absorbance at 510 nm plotted against the time after irradiation (black circles). The data were fitted by single exponential function $y=y_0-b \times \exp(-t/\tau)$ (solid curve, $y=0.029-0.026 \times \exp(t/10500)$, $\tau=1.1 \times 10^4$ sec).” (page 34, lines 12–21)

Reviewers' Comments:

Reviewer #3:

Remarks to the Author:

The authors answered my questions to full satisfaction.

Reviewer #4:

Remarks to the Author:

Overall I am happy with the authors' response to my comments. This is impressive work. My only suggestion is that the authors may wish to return to the issue of the likely in vivo photoactivation rate of Opn5L. Their new data on this topic addresses the efficiency of the pigment, which is appropriate as this is what they can measure. However, it should also be possible to provide a rough estimate of the amount of pigment in any given cell (based upon estimates of pigment in vertebrate rods and the difference in cell surface area), and the photon flux in the target brain areas for typical environmental lighting conditions (based upon estimates of transmission of bird tissues). From these two types of information one could estimate the potential rate of photoresponse. The numbers would not be perfect, but it would be good to know whether its likely to be 10 events per second or 1 event per day for physiologically sensible parameters.

Reviewer #5:

Remarks to the Author:

The revisions made to the manuscript clearly and fully address the comments raised in my report on the original version. The revised version is an outstanding contribution, and I recommend its acceptance for publication by NCOMMS. One minor stylistic point: In the Supplementary Information section (p. 11, line 147 of this section), the word "simulated" might be confusing to some readers. It would be clearer if the text instead read something like "Single exponential functions fitted to the recovery data exhibited a time constant....".

Comment from Reviewer 4:

My only suggestion is that the authors may wish to return to the issue of the likely in vivo photoactivation rate of Opn5L.

Response:

According to your comment, we estimated the numbers of Opn5L1 molecules photobleached in PVN, the innermost part of the brain where Opn5L1 is present, under sunlight. For this purpose, we referred to a radius of the large neural cells present in PVN (10 μm) (Kiss et al., J. Comp. Neurol. (1991)) and the density of melanopsin in retinal ganglion cells (Do et al. Nature (2009)). We also assumed that the fraction of sunlight reaching to the base of avian brain ($0.2\text{-}2.0 \times 10^{12}$ photons $\cdot\text{m}^{-2}\cdot\text{s}^{-1}\cdot\text{nm}^{-1}$ (370-600 nm)) (Foster & Follett J. Comp. Physiol. (1985)) can be applicable in the present study. Then using molecular extinction coefficient and quantum yield that we determined in the present study, we estimated that one Opn5L1 molecule is photobleached about every 8 minutes in one cell. If the density of Opn5L1 is similar to that of rhodopsin in the disk membranes of rod photoreceptor cells (Pugh & Lamb Biochim. Biophys. Acta. (1993)), about 20 molecules of Opn5L1 are bleached in 1 second. In these estimations, we assumed that Opn5L1 is distributed only in somatic membrane. If Opn5L1 is expressed also in dendritic membrane, more Opn5L1 molecules would be photobleached. These estimations may support our hypothesis that Opn5L1 works as a reverse photoreceptor even in the hypothalamus. We added the following sentences in the Discussion section of the revised manuscript'

“Based on the intensity of the solar light reaching the base of brain ($0.2-2.0 \times 10^{12}$ photons $m^{-2} s^{-1} nm^{-1}$ (370-600 nm))⁴³ and the size of cells in PVN (10 μm radius)⁴⁴, one can estimate the rate of photoresponse of Opn5L1 in one neuron to be one per several minutes, assuming that the density of Opn5L1 is equivalent to that of melanopsin in retinal ganglion cells ($3 \mu m^{-2}$)⁴⁵. If the density of Opn5L1 is equivalent to that of rhodopsin in the disk membranes of rod photoreceptor cells ($25,000 \mu m^{-2}$)⁴⁶, the estimate rise to about 20 molecules of Opn5L1 in 1 second.”

Comment from Reviewer 5:

One minor stylistic point: In the Supplementary Information section (p. 11, line 147 of this section), the word “simulated” might be confusing to some readers. It would be clearer if the text instead read something like “Single exponential functions fitted to the recovery data exhibited a time constant....”.

Response

According to your comment, we revised the sentence in p. 11, line 147 as follows:

“Single exponential functions fitted to the recovery data exhibited time constants of 2.57 (red broken curve) and 2.33 (black solid curve) hours, respectively.”